

# SA3C-ID: a novel network intrusion detection model using feature selection and adversarial training

Wanwei Huang[1], Haobin Tian[1], Lei Wang[2], Sunan Wang[3], Kun Wang[4] and Songze Li[5]

[1] College of Software Engineering, Zhengzhou University of Light Industry, Zhengzhou, Henan, China
[2] China Research Institute of Radio Wave Propagation, Xinxiang, Henan, China
[3] Shenzhen Polytechnic University, Shenzhen, Guangdong, China
[4] Zhengzhou Xinda Institute of Advanced Technology, Zhengzhou, Henan, China
[5] Henan Xindawangyu Science & Technology Co., Ltd., Zhengzhou, Henan, China

Corresponding author
Sunan Wang,
wangsunansps@163.com

## ABSTRACT

With the continuous proliferation of emerging technologies such as cloud computing, 5G networks, and the Internet of Things, the field of cybersecurity is facing an increasing number of complex challenges. Network intrusion detection systems, as a fundamental part of network security, have become increasingly significant. However, traditional intrusion detection methods exhibit several limitations, including insufficient feature extraction from network data, high model complexity, and data imbalance, which result in issues like low detection efficiency, as well as frequent false positives and missed alarms. To address the above issues, this article proposed an adversarial intrusion detection model (Soft Adversarial Asynchronous Actor-Critic Intrusion Detection, SA3C-ID) based on reinforcement learning. Firstly, the raw dataset is preprocessed *via* one-hot encoding and standardization. Subsequently, the refined data undergoes feature selection employing an improved pigeon-inspired optimizer (PIO) algorithm. This operation eliminates redundant and irrelevant features, consequently reducing data dimensionality while maintaining critical information. Next, the network intrusion detection process is modeled as a Markov decision process and integrated with the Soft Actor-Critic (SAC) reinforcement learning algorithm, with a view to constructing agents; In the context of adversarial training, two agents, designated as the attacker and the defender, are defined to perform asynchronous adversarial training. During this training process, both agents calculate the reward value, update their respective strategies, and transfer parameters based on the classification results. Finally, to verify the robustness and generalization ability of the SA3C-ID model, ablation experiments and comparative evaluations are conducted on two benchmark datasets, NSL-KDD and CSE-CIC-IDS2018. The experimental results demonstrate that SA3C-ID exhibits superior performance in comparison to other prevalent intrusion detection models. The F1-score attained by SA3C-ID was 92.58% and 98.76% on the NSL-KDD and CSE-CIC-IDS2018 datasets, respectively.

## INTRODUCTION

Recent advancements in information technology have accelerated the integration of emerging technologies like cloud computing, 5G networks, and IoT into daily life. While these innovations enhance societal intelligence and usher in technological sophistication, they also create unintended consequences. Specifically, there has been a sharp surge in the frequency of cyber-attacks, and the methods employed have become increasingly diversified and complex (*Duo, Zhou & Abusorrah, 2022*). This has resulted in a disruption of the normal order and stable operation of cyberspace and poses a serious threat to its security and stability. To address network security challenges, the most effective approach is to deploy an intrusion detection system (IDS) on computers (*Dina & Manivannan, 2021*).

An intrusion detection system is a proactive security technology that protects networks or hosts from potential threats. It operates by scanning systems for signs of intrusion activities or security vulnerabilities. The detection process involves identifying either the characteristics of known attacks or deviations from normal activity patterns. Based on their detection techniques, intrusion detection systems can be categorized into two main types: misuse detection and anomaly detection (*Sarker et al., 2023*). Misuse detection (*Radivilova et al., 2020*) is a knowledge-based approach, where the detection system first needs to explicitly define the behavioral characteristics of the intrusion (*Bronte, Shahriar & Haddad, 2016*) and identify the attack by matching the features or rules. Anomaly detection is a behavior-based approach that involves the initial establishment of the subject's normal activity patterns. This is followed by the subsequent determination of any deviations in behavior that diverge from the established baseline. Anomaly detection (*Altwaijry, ALQahtani & AlTuraiki, 2020*) is a process that requires defining the normal state of the subject. It does not require prior knowledge of intrusion behavior and can detect novel, previously unknown attacks.

The network intrusion detection system (NIDS) is a critical component in the defense of network security. It is responsible for the real-time monitoring of network traffic, the accurate identification of abnormal behavior, and the rapid response to intrusion incidents (*He, Kim & Asghar, 2023*). It is an essential component for the safeguarding of network security and the maintenance of network stability.

Conventional intrusion detection systems exhibit three critical limitations in complex network environments, directly impeding their effectiveness in modern cybersecurity scenarios. First, high-dimensional network traffic data contains numerous irrelevant and redundant features, which traditional methods fail to efficiently filter. This not only extends model training time (*Ganapathy et al., 2013*) but also degrades prediction accuracy by overwhelming classifiers with noise (*Chen et al., 2023*). Second, severe data imbalance skews model learning, as normal traffic vastly outnumbers rare attack instances (*Yi et al., 2023*). Existing models often prioritize majority-class normal patterns, leading to poor detection of minority attacks that pose significant risks but lack sufficient training samples (*Zhang & Liu, 2022*). Third, these systems suffer from architectural complexity when processing large-scale data. Their computationally intensive nature leads to inefficient

training and delayed detection responses, failing to satisfy real-time security demands (*Thakkar & Lohiya, 2022*).

To effectively address the above challenges, this article introduces Soft Adversarial Asynchronous Actor-Critic Intrusion Detection (SA3C-ID), an asynchronous adversarial intrusion detection model that combines the simulated annealing-based binary pigeon-inspired optimizer (SABPIO) for feature selection with soft actor-critic reinforcement learning. By separating the detection process into feature selection and adversarial training stages, SA3C-ID outperforms the state-of-the-art AE-RL model on the NSL-KDD dataset, boosting the F1-score by 16.6%. Additionally, it cuts training time by 53% and improves the detection of minority-class attacks, thereby strengthening the security and stability of network environments. The main contributions of this article are as follows:

(1) Investigates the state-of-the-art in cyberspace security feature selection and intrusion detection algorithms, introducing SA3C-ID, an asynchronous adversarial model that integrates the SABPIO with SAC reinforcement learning;

(2) Addressing network traffic imbalance and variable attack category proportions, a class-weighted adaptive reward function is designed. This leverages attacker-defender interaction results in complex scenarios to prioritize minority-class attack detection, improving model sensitivity to rare intrusions;

(3) Decomposes network intrusion detection into feature selection and adversarial detection stages. Introduces an asynchronous adversarial training mechanism to simulate real attack-defense dynamics, enabling the model to rapidly detect and respond to complex, evolving network attacks;

(4) Conducts ablation experiments on NSL-KDD and CSE-CIC-IDS2018 to validate the effectiveness of key modules. Performs comparative analysis with advanced models to demonstrate SA3C-ID's superior performance in detection accuracy and efficiency.

The remainder of this article is structured as follows: "Related Work" provides an overview of related work previously conducted by other researchers; "SA3C-ID Model" introduces the overall architecture, core modules, and algorithm flow of the SA3C-ID model; "Experimental Results and Discussion" presents simulated ablation experiment, comparative experiment, performance evaluation, and result discussion on SA3C-ID model; and "Conclusion" concludes and discusses future work.

## RELATED WORK

In the contemporary network environment, characterized by its complexity and constant evolution, the ability to respond quickly and effectively to network attacks has emerged as a pivotal research imperative for ensuring network security. Among the array of security technologies available, network intrusion detection has garnered significant attention from researchers both within and outside national borders, owing to its strong detection capabilities.

| Table 1 Characteristics of feature selection algorithms. | | |
|---|---|---|
| **Feature selection algorithm** | **Advantages** | **Disadvantages** |
| Filter-based | Faster computation; provides feature importance evaluation. | May lose some useful feature information due to ignoring model-specific interactions. |
| Embedded-based | Performs feature selection during model training, avoiding inconsistency between feature selection and classification. | Requires integration with specific machine learning algorithms; higher computational cost for large datasets. |
| Wrapper-based | Generally higher accuracy than filter-based and embedded-based methods. | Higher computational cost; sensitive to the size and quality of training data. |

## Feature selection algorithm in NIDS

In complex network environments, challenges such as large traffic data volumes and high data dimensionality significantly constrain model classification performance. In response to the increasing data volume, researchers have investigated various sample selection methods to optimize the training process. At the same time, feature selection techniques have been developed to address issues such as high data dimensionality and irrelevant or redundant features (*Di Mauro et al., 2021*). Table 1 provides a detailed comparison of the key feature selection algorithms, outlining their respective advantages and disadvantages to offer a clearer understanding of their applicability in tackling network data complexities.

*Halim et al. (2021)* proposed GbFS, a feature selection method based on genetic algorithms (GA), to address the challenges of high-dimensional data on learning algorithm performance and the high false positive and false negative rates in existing IDSs. They developed a novel GA fitness function, utilizing a calculation of the relevance of selected features without class labels. The experimental results obtained demonstrated that GbFS can significantly improve classifier accuracy, reaching up to 99.80%.

*Nazir & Khan (2021)* proposed a feature selection method based on Tabu Search-Random Forest (TS-RF), utilizing Tabu Search for feature selection and Random Forest as the learning algorithm. They designed a fitness function incorporating classification accuracy, false positive rate, and the number of features. The experimental results, which were obtained by comparing TS-RF with traditional and existing feature selection methods on the UNSW-NB15 dataset, demonstrate that TS-RF significantly reduces both the number of features and time complexity.

*Bhattacharya et al. (2020)* proposed an XGBoost classification model based on the principal component analysis (PCA)-Firefly algorithm. The initial step involved the transformation of the dataset, collected from Kaggle, into the appropriate format using One-Hot encoding. The PCA-Firefly algorithm was then applied to extract low-dimensional key information from high-dimensional data, and to measure the attractiveness of feature subsets by calculating the brightness of the fireflies and the distance between them. Finally, the XGBoost algorithm was employed for classification. The experimental results demonstrate that, in comparison with traditional machine learning algorithms, this model outperforms in terms of accuracy, sensitivity, and specificity, thereby enhancing the performance of intrusion detection.

*U & Kumar (2024)* proposed a network intrusion detection method based on the enhanced whale optimization algorithm (EWOA) and integrated classifiers. First, linear discriminant analysis (LDA) and EWOA were used to reduce the data dimensionality, followed by classification with Random Forest and XGBoost integrated classifiers. EWOA effectively addressed the shortcomings of the WOA, including the imbalance between exploration and exploitation, slow convergence speed, and low efficiency in high-dimensional spaces. Combined with LDA, this improved feature selection efficiency. The experiments used the NSL-KDD dataset, and the results showed that the overall classification accuracy of the system reached 96.08%.

*Ammannamma & Chakravarthy (2025)* proposed a pioneering intrusion detection method that combines bio-inspired feature selection with a convolutional Ghost-Net deep capsule network. For feature selection, the authors developed an enhanced version of the gazelle optimization algorithm (Mod-GO) to select the most relevant features for the model. This approach enhances the effectiveness of the intrusion detection model, reduces the false alarm rate, and addresses the issue of time complexity.

Existing feature selection methods in NIDS, like GbFS, TS-RF, PCA-Firefly, EWOA, and Mod-GO, have limitations. They may suffer from slow convergence, getting trapped in local optima, loss of important features, and poor performance in imbalanced datasets. The SA3C-ID model's SABPIO addresses these by combining pigeon-inspired optimization with simulated annealing. It expands the search space, avoids local optima, retains relevant features, and adapts to imbalanced data, outperforming traditional methods.

## Deep learning-based NIDS

Deep learning has shown vigorous development in academic research and industrial practice in recent years, attracting much attention from academia and the industry, and has been widely used in the field of intrusion detection. The focus of network intrusion detection based on deep learning is the identification of intrinsic patterns and representation levels within data, with the extraction of features achieved through the construction of multi-layer neural networks. Research has demonstrated that deep learning-based intrusion detection methods generally exhibit superior performance in comparison to traditional approaches.

*Caville et al. (2022)* proposed Anomal-E, an anomaly detection method based on graph neural network (GNN). The method first preprocesses the original network stream to generate training and test graphs, and the training graph is then used to train the E-graphSAGE encoder, which learns edge representations and adjusts parameters by manipulating edge features in a process similar to DGI. Following the training phase, the output graph embedding is then utilized to train four anomaly detection algorithms. The experimental results, which were based on two benchmark datasets, demonstrate a significant performance improvement of Anomal-E in comparison to other baseline algorithms.

*Kasongo (2023)* proposed an intrusion detection system (IDS) framework based on recurrent neural networks (RNNs). The framework utilizes different types of RNNs

combined with XGBoost for feature selection to extract key features and perform classification. Experimental results on the NSL-KDD and UNSW-NB15 datasets demonstrate that the proposed IDS framework outperforms various models across different classification tasks and achieves a shorter training time.

*Nandanwar & Katarya (2024)* proposed AttackNet, a robust deep learning model based on an adaptive convolutional neural networks-gated recurrent unit (CNN-GRU) architecture. The model optimizes data through preprocessing techniques such as data augmentation, shuffling, encoding, and standardization. It uses CNNs to extract spatial features and GRUs to capture temporal dependencies, enabling the detection and classification of various botnet attacks in IIoT. Experimental results show that AttackNet performs well in detecting different types of attacks, achieving higher accuracy than other methods, and demonstrating strong reliability, trustworthiness, and scalability.

*Aljehane et al. (2024)* proposed an intrusion detection system based on the golden jackal optimization algorithm with deep learning (GJOADL-IDSNS). The process starts with data normalization, followed by feature selection using the GJOA algorithm, which simulates the hunting behavior of golden jackals. Network intrusion detection is then performed using the A-BiLSTM model with an attention mechanism, and the model hyperparameters are fine-tuned using SSA. Experimental results on the CICIDS-2017 dataset show that the proposed IDS outperforms CNN, LSTM and other models in intrusion classification tasks.

*Devendiran & Turukmane (2024)* proposed a deep learning-based Dugat-LSTM intrusion detection system, which involves data cleaning, M-squared normalization, extended synthetic sampling, KerPCA feature extraction, and CHbO feature selection. The Dugat-LSTM model achieves an accuracy of 98.76% and 99.65% on the TON-IOT and NSL-KDD datasets, respectively.

*Hazman et al. (2024)* propose IDS-SIoDL, a deep learning-based intrusion detection system for IoT smart cities, integrating LSTM and feature engineering. Using autoencoders and genetic algorithms for feature optimization, it achieves 99.94% accuracy on BoT-IoT, nearly 100% on Edge-IIoT, and 99.70% on NSL-KDD, with efficient training (600 ms) and classification (6 ms).

*Nazir et al. (2024)* propose a novel deep learning-based hybrid CNN-LSTM architecture for efficient IoT threat detection, integrating CNN's spatial feature extraction with LSTM's temporal modeling. Using PCA for data optimization and techniques like model quantization/pruning for resource-constrained deployment, the model achieves 95% accuracy on IoT-23 and 99% on N-BaIoT and CICIDS2017 datasets, demonstrating strong adaptability to diverse IoT environments.

Deep learning-based NIDS models, such as Anomal-E and RNN-XGBoost, outperform traditional ones in feature processing but have flaws. They suffer from high computational complexity, poor generalization to new attacks, and ineffective handling of class imbalance. SA3C-ID tackles these problems. Its SABPIO module simplifies data through feature selection, cutting down computational load. By using MDP and SAC, it adapts to imbalanced data *via* a tailored reward function and improves detection efficiency with asynchronous training and priority experience replay.

## Reinforcement learning-based NIDS

Reinforcement learning (RL) is used to describe and solve the problem of maximizing rewards or achieving specific goals by learning a strategy through an agent's interaction with a dynamic environment. Reinforcement learning has been extensively utilized in domains such as robot control and industrial manufacturing, facilitating dynamic decision-making through the gradual development of strategies through the interaction between the agent and the environment. However, research on reinforcement learning in network intrusion detection remains in its infancy.

*Lopez-Martin, Carro & Sanchez-Esguevillas (2020)* proposed a deep reinforcement learning-based intrusion detection system that uses DRL algorithms such as DQN, DDQN, PG, and AC. They applied these algorithms to intrusion detection using labeled datasets, treating network features as states and labeled values as actions to prepare the dataset and train the model. Experimental results on the NSL-KDD and AWID datasets demonstrated that DRL outperforms existing machine learning methods in improving intrusion detection results through model and parameter tuning.

*Tellache et al. (2024)* proposed an intrusion detection system (IDS) based on multi-agent reinforcement learning. This IDS utilized a two-layer framework consisting of multiple agents specialized in detecting specific attacks and a decision-making agent. The authors improved the deep Q-network (DQN) algorithm by incorporating a weighted mean square loss function and cost-sensitive learning techniques. Experimental findings on the CIC-IDS-2017 dataset demonstrated the efficacy of this approach in addressing the class imbalance problem, while concurrently achieving fine-grained attack classification with a minimal false alarm rate.

*Ren et al. (2022)* proposed an ID-RDRL intrusion detection system based on deep reinforcement learning. This system utilizes RFE and DT to first screen the ideal subset of features that best reflect the deep information of the original data, and these features are then fed into the neural network. The classifier is then trained to identify network intrusions using the DQN algorithm. The experimental results on the CSE-CIC-IDS2018 dataset demonstrate that this method can effectively remove redundant features and better identify various kinds of attacks than MLP, CNN, and other models.

*Caminero, Lopez-Martin & Carro (2019)* proposed an AE-RL intrusion detection system based on adversarial reinforcement learning. AE-RL integrates reinforcement learning with supervised learning frameworks, enabling the environment agent and the classifier agent to be trained in parallel based on adversarial strategies, and uses two different Q functions to achieve dynamic intelligent resampling to alleviate the problem of dataset imbalance. Experimental results on the NSL-KDD and AWID datasets demonstrate the efficacy of AE-RL in addressing class imbalance, exhibiting reduced training and prediction times while maintaining a high detection capability for less prevalent attack categories.

*Li et al. (2023)* proposed an AE-SAC intrusion detection model based on adversarial environment and SAC reinforcement learning algorithm. The model introduces an environmental agent to resample the original training data, and adopts the SAC algorithm

**Table 2 Summary of related works.**

| Method | Core | Defects |
|---|---|---|
| Anomal-E | E-GraphSAGE encoder + modified DGI; edge embeddings fed to anomaly detectors (PCA, IF, *etc.*) | Complex preprocessing; real-time efficiency unaddressed. |
| RNN-XGBoost | RNN variants (LSTM/GRU) + XGBoost for feature selection/classification. | Limited high-dimensional heterogeneous data handling. |
| Adaptive CNN-GRU | 1D CNN (spatial) + GRU (temporal) + data preprocessing + softmax. | High computational load from deep architecture. |
| GJOADL-IDSNS | GJOA for feature selection, Attention GJOA (feature selection) + A-BiLSTM (classification) + SSA (hyperparameter tuning). | High computational overhead from optimization algorithms. |
| Dugat-LSTM | Multi-step preprocessing (M-squared, KerPCA/CHbO) + LSTM classification. | Limited generalization to unknown attacks. |
| IDS-SIoDL | LSTM + feature engineering (Autoencoder, GA, IG) for preprocessing/classification. | Unvalidated generalization to unseen attack variants. |
| Hybrid CNN-LSTM | CNN (spatial) + LSTM (temporal) + PCA optimization + model pruning. | Limited unknown/novel attack detection. |
| DRL-based IDS (DQN, DDQN, PG, AC) | DRL treats network features as states, labels as actions. | High training complexity |
| MARL-based IDS | Two-layer architecture (detection agents + decision agent) + improved DQN. | Increased system complexity. |
| ID-RDRL | RFE + DT (feature selection) + DQN classification. | Poor non-linear feature handling. |
| AE-RL | Adversarial training of environment/classifier agents; dynamic resampling *via* Q-functions. | Stability issues in parallel training. |
| AE-SAC | Environmental agent for data resampling and SAC algorithm to maximize action entropy and cumulative reward. | Complex structure causing longer training time. |
| Big-IDS | Decentralized MARL + shared target networks + cloud/streaming techniques. | High computational cost (encryption/distributed training). |
| RFS-DRL | DQN + RFE (feature selection) + epsilon-greedy/experience replay. | Low U2R detection accuracy; high hardware requirements. |

to maximize the action entropy of the policy output while maximizing the cumulative reward expectation. AE-SAC achieved an F1-score of 83.97% and 98.9% on the NSLKDD and AWID datasets, respectively, and the trained model structure is simple and easy to deploy.

*Louati, Ktata & Amous (2024)* propose Big-IDS, a decentralized multi-agent reinforcement learning (MARL) intrusion detection system for big data networks, integrating cloud computing and streaming techniques. Using shared target networks and encrypted agent communication, it achieves 97.44% accuracy on NSL-KDD, enabling real-time detection and efficient handling of large-scale data.

*Sharma & Singh (2024)* propose a batch reinforcement learning-based intrusion detection model, integrating deep Q-network with recursive feature elimination (RFE). Using correlation, information gain, and RFE for feature selection, the model achieves 99.12% accuracy and 99.2% F1-score on NSL-KDD, excelling in detecting low-frequency attacks while reducing computational load.

RL-based NIDS, like DQN and Big-IDS, show potential in threat detection but face issues. High training complexity, unstable asynchronous training, and improper handling of

of state-action space in imbalanced datasets are their main problems. SA3C-ID offers a solution. It uses an asynchronous adversarial training mechanism with two agents and a dynamic reward function. The SAC algorithm is applied to explore the state-action space, and asynchronous updates optimize resource use. This helps in better handling of class imbalance and faster convergence, enabling more precise threat detection.

To provide a comprehensive overview and facilitate a clear comparison, Table 2 summarizes the key aspects of related works, including their core architectures and notable defects. This table serves as a crucial reference for understanding the current state-of-the-art in the field and identifying areas for improvement, thereby highlighting the significance of our proposed method in overcoming these existing drawbacks.

## SA3C-ID MODEL

This section introduces the intrusion detection model SA3C-ID (Soft Adversarial Asynchronous Actor-Critic Intrusion Detection) proposed in this study and outlines the structure of the model. The SA3C-ID model integrates three core modules to address IDS challenges: (1) SABPIO reduces feature dimensionality to accelerate training; (2) asynchronous adversarial training simulates real attack-defense dynamics; (3) a class-weighted reward function prioritizes rare attacks. These components work synergistically to enhance detection efficiency and accuracy.

The framework structure of the SA3C-ID model is shown in Fig. 1, which is divided into a data preprocessing module, a SABPIO feature selection module, a Mini-Batch data generation module, and a SA3C adversarial intrusion detection module. First, the dataset is preprocessed by means of one-hot encoding and standardization, and then the processed data is input to the feature selection section; Second, the SABPIO algorithm is used to select features for the dataset, eliminating redundant and irrelevant features and thus reducing the dimensionality of the data; Third, the Mini-Batch training strategy is used to re-encode the data processed based on the optimal feature subset, and the re-encoded data is input into the SA3C model; Finally, the SAC reinforcement learning algorithm is combined to build an agent, define the adversarial agents of the attacker and defender, conduct asynchronous adversarial training, and build a classifier for the defender agent to classify the traffic. The specific implementation process is as follows:

(1) The dataset is first subjected to a series of preprocessing steps, including data integration, cleansing, transformation, and normalization. And then transferred to the feature selection stage to identify the desired subset of features;

(2) Feature selection using the SABPIO feature selection algorithm, which explores the potential optimal solution region in the feature space by simulating the migratory behavior of pigeon flocks, searches for the optimal subset of features, and eliminates redundant and irrelevant features;

(3) The soft adversarial asynchronous actor-critic (SA3C) algorithm has been proposed as a means of enhancing the efficacy of the SAC algorithm. The approach involves the construction of a neural network model, the initialization of parameters, and the mapping of preprocessed data into states, actions, and reward values. This is then

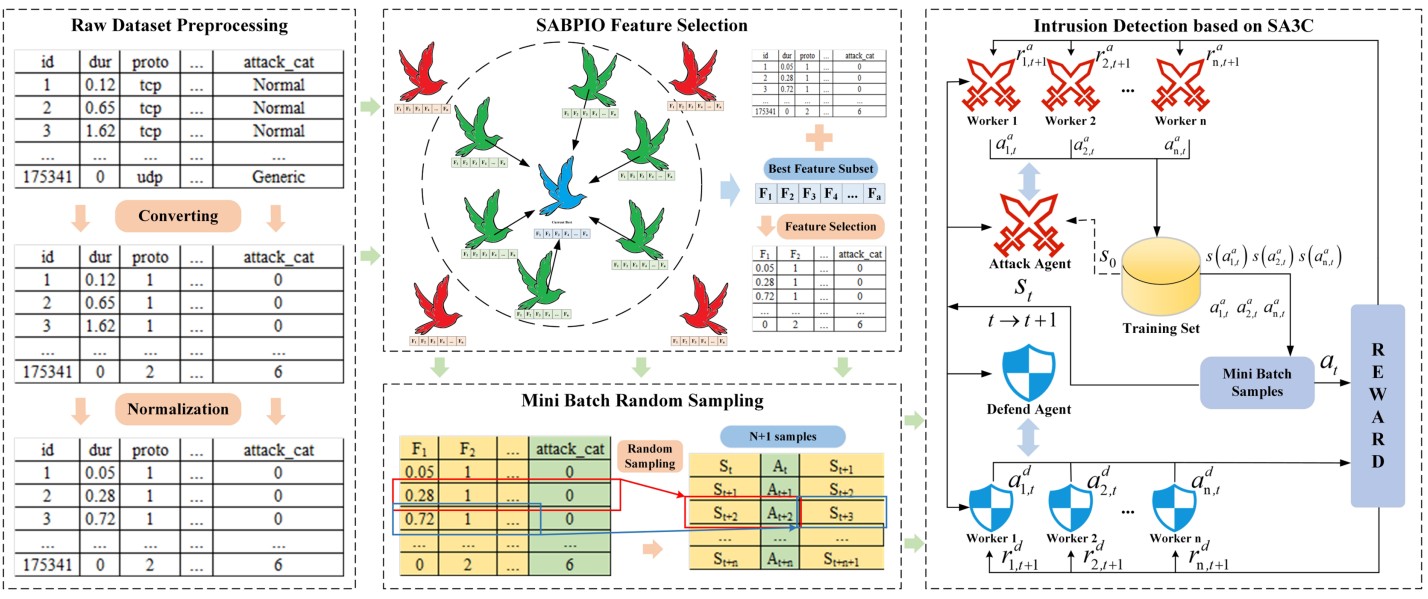

**Figure 1** SA3C-ID model framework.

combined with the utilization of experience replay pool technology, thereby enabling the stable and efficient optimization of network parameters, as well as the efficient utilization of computing resources during the training process through asynchronous updates;

(4) Introducing an adversarial training mechanism, defining two intelligences, the attacker and the defender, for iterative confrontation, simulating real attack and defense scenarios, and both attackers and defenders continue to optimize their classification strategies through reinforcement learning.

## Data preprocessing module

(1) Integration

In the dataset used in this article, the NSL-KDD dataset has been integrated and divided into training and test sets at the time of release, while the CSE-CIC-IDS2018 dataset needs to be downloaded, organized, and divided by the users themselves. The CSE-CIC-IDS2018 dataset contains normal network traffic data and seven types of network attack data, totaling 10 CSV files. Each file contains network traffic data for different time periods. To ensure the integrity and consistency of the data, it is imperative to unify the CSV files through a merge operation and to format them in accordance with the stipulated requirements of the dataset.

(2) Cleaning

With regard to the NSL-KDD and CSE-CIC-IDS2018 datasets addressed in this article, the primary focus of data cleaning is the removal of duplicates and the processing of outliers and missing values. In the NSL-KDD dataset, although pre-processing has been performed prior to release, further verification is still required to ensure that no duplicate records are overlooked. For the self-organized CSE-CIC-IDS2018 dataset, duplicate data

may be introduced when merging CSV files, and it is necessary to remove duplicates based on unique identifiers. At the same time, for outliers and missing values that may appear in the dataset, deletion, filling, or interpolation methods are used according to the specific situation to ensure the integrity and accuracy of the dataset and the effectiveness of subsequent analysis.

(3) Conversion

During the execution of the intrusion detection task by the model, numerical data is accepted for training and testing purposes; therefore, it is necessary to transform non-numerical data in the dataset into numerical data. As a case in point, the NSL-KDD dataset features three non-numeric features among its 41 feature attributes, which are converted using one-hot encoding. To illustrate, the protocol_type feature type features the values "tcp", "udp" and "icmp", which are converted into numerical data post-digitization. The same method is used to convert the service feature and flag feature into numerical data.

(4) Normalization

Normalization is a data preprocessing technique that facilitates the comparison and analysis of data by transforming data of differing magnitudes and distributions into a common scale. Normalization has been demonstrated to enhance the accuracy and efficiency of data analysis and machine learning algorithms by mitigating bias resulting from variations in variables. The normalization method is shown in Eq. (1).

$$x_{norm} = \frac{x - x_{min}}{x_{max} - x_{min}}. \tag{1}$$

In the above equation, $x_{max}$ is the maximum value of the eigenvalue, $x_{min}$ is the minimum value of the eigenvalue, $x_{norm}$ is the output value, the output value is between [0, 1].

## SABPIO feature selection module

To address the high-dimensional feature redundancy challenge in IDS, this article uses an improved binary pigeon inspired optimizer algorithm (SABPIO) (*Huang et al., 2024*) for feature selection, and the algorithm flow is shown in Fig. 2. This approach uses mutation and simulated annealing during the map and compass stage to expand the search scope and prevent the feature subset from being trapped in local optima. It also introduces a population decay factor in the landmark stage to control rapid population decline and regulate the algorithm's convergence rate. For a more in-depth understanding of the algorithm's structure, parameter settings, and theoretical foundation, please refer to our previous work (*Huang et al., 2024*), where the algorithm is comprehensively described from both theoretical derivation and practical implementation perspectives.

### Map and compass operator stage

The SABPIO algorithm introduces a simulated annealing mechanism, a multidimensional similarity strategy, and a mutation mechanism in the map and compass operator stage to address the issue of traditional PIO algorithms converging too quickly and falling into local optima too early.

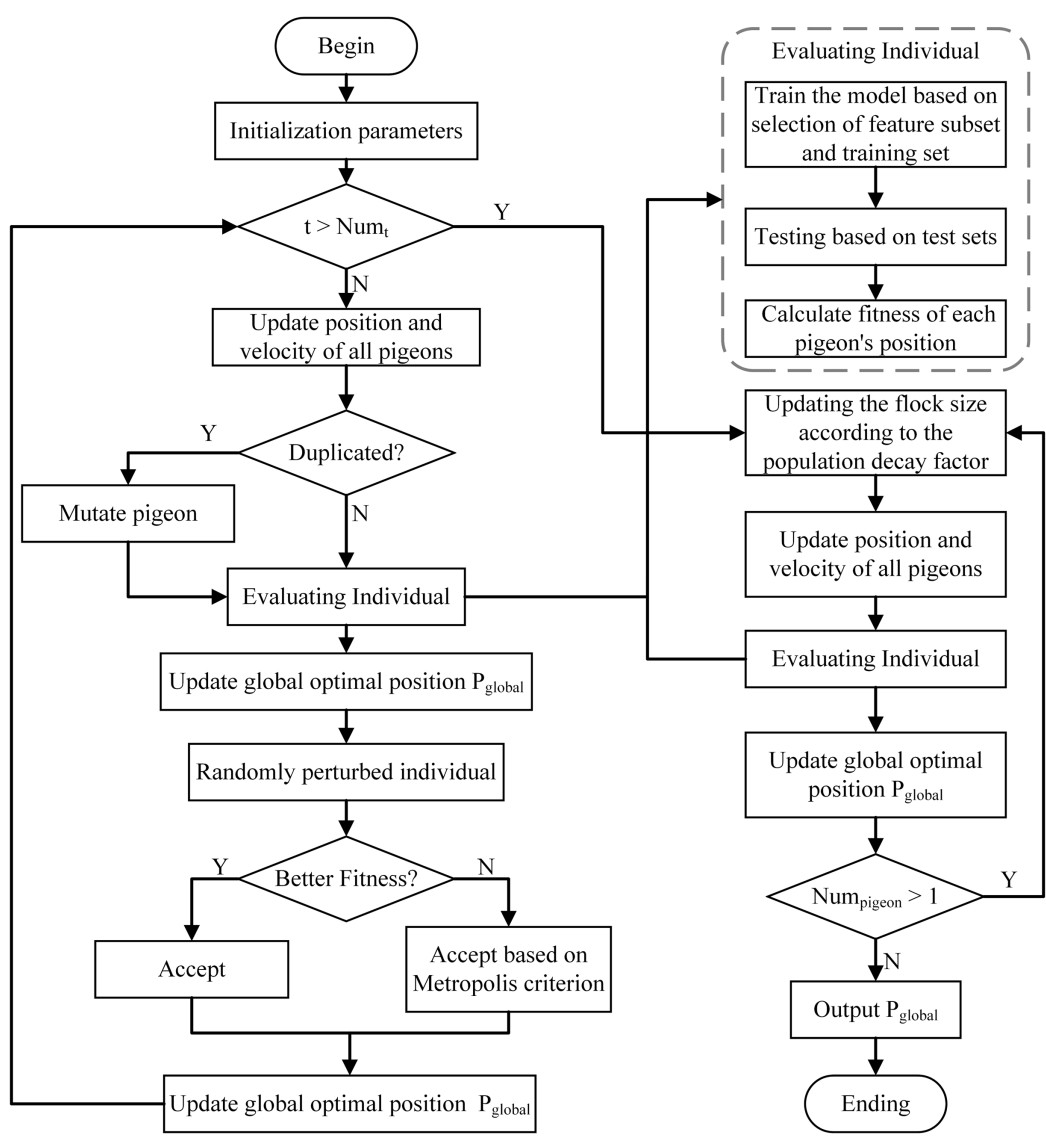

**Figure 2  Flow of the SABPIO algorithm.**

In each iteration of the SABPIO algorithm, an inner loop based on the simulated annealing algorithm is appended to the map and compass operator stage. Within this loop, each value in the random feature subset is subjected to random perturbation, after which the fitness is recalculated. The new feature subset is accepted probabilistically using the Metropolis criterion, as demonstrated in Eq. (2).

$$p\left(P_{global} \Rightarrow P_i'\right) = \begin{cases} 1 & \Delta E < 0 \\ e^{-\frac{\Delta E}{T}} & otherwise \end{cases} \tag{2}$$

as where $p\left(P_{global} \Rightarrow P_i'\right)$ denotes the probability of accepting the new solution $P_i'$, and $\Delta E$ denotes the energy difference (*i.e.*, the difference in fitness between the old and new solutions).

When updating the position of the pigeon flock, the SABPIO algorithm proposed a position update formula derived from the Pearson correlation coefficient, cosine similarity, and Jaccard similarity coefficient. Utilizing weighted calculation enables the mitigation of limitations associated with the measurement method, thereby circumventing the defects inherent to a solitary measurement method, as illustrated in Eq. (3).

$$
\begin{aligned}
V_i &= CompositeSim(P_i, P_{global}) \\
&= \omega_1 Pearson(P_i, P_{global}) + \omega_2 Consine(P_i, P_{global}) + \omega_3 [2Jaccard(P_i, P_{global}) - 1].
\end{aligned} \tag{3}
$$

The utilization of the twofold Jaccard correlation coefficient minus one in the formula is intended to reduce the domain of values to an equivalent range as the Pearson correlation coefficient and cosine similarity. $\omega_1$, $\omega_2$ and $\omega_3$ are weighting factors and $\omega_1 + \omega_2 + \omega_3 = 1$.

In order to circumvent the occurrence of duplicate solutions, which have been demonstrated to diminish the efficacy of the algorithm's search capability, the SABPIO algorithm has been engineered to ascertain the presence of any existing solutions prior to the process of updating a given solution. In the event that a duplicate solution is identified, the SABPIO algorithm employs a process of random mutation, modifying each dimension of the solution according to a uniform random number. This approach serves to augment the search space, thereby enhancing the algorithm's capacity to explore a more extensive range of possibilities.

### Landmark operator stage

In the landmark operator stage, all solutions are sorted according to their fitness values in each iteration. Solutions with lower fitness values are then eliminated. SABPIO proposed the population decay factor $\alpha$ to control the decay rate of the population in this stage. The population decay is shown in Eqs. (4) and (5).

$$
\alpha = \beta \cdot e^{-\dfrac{t}{\log_2 Num_{pigeon}}} \tag{4}
$$

$$
Num_{pigeon}^t = \alpha \cdot sort(Num_{pigeon}^{t-1}). \tag{5}
$$

In Eq. (4), $\beta$ is defined as the interval between 0 and 1, and $t$ is the number of landmark operator iterations. The SABPIO algorithm improves the traditional pigeon inspired optimizer algorithm by halving the decay with each iteration, and updates the population size according to Eq. (5) to prevent rapid population decay.

## SA3C-ID detection model

### Problem mapping

This article models the detection process of network attack behavior as a Markov decision process (MDP). The modeled MDP is defined as a four-tuple $M = <S, A, R, \gamma>$, where $S$ is the state space, $A$ is the action space, $R$ is the reward space, and $\gamma$ is the reward discount factor. In standard MDPs, the state transition probability $P(s'|s, a)$ is typically defined as the likelihood of the environment transitioning to the next state $s'$ given a state $s$ and an action $a$. However, in this article, the state transition probability is not explicitly modeled.

The SAC algorithm is a model-free method that utilizes reinforcement learning. It obtains state transition experience through interaction with the environment and uses experience replay to update the strategy and value function. The algorithm adaptively learns the optimal strategy for attack detection without explicitly modeling the dynamic characteristics of the environment.

For the state space $S(t) = \{s_1, s_2 \ldots s_t\}$, we consider all data features in the network traffic data except labels as the current state $s_t$, including various statistics and metrics of the network traffic, such as packet sizes, protocol types, traffic rates, *etc.* Each state $s_t$ in $S(t)$ is a feature vector that describes the current situation of the network traffic.

In each state $s_t$, the agent samples the probability distribution of the action output by the action strategy function $\pi^*(a|s)$ to obtain the specific action, that is, the label of the network traffic behavior, to form the action space $A(t) = \{a_1, a_2 \ldots a_t\}$. The calculation formulas are shown in Eqs. (6) and (7).

$$\pi^*(a|s) = \operatorname*{arcmax}_{\pi} E_\pi \left[ \sum_{t=0}^{T} \gamma^t (r(s_t, a_t) + \alpha H(\pi(\cdot|s_t))) \right] \tag{6}$$

$$H(\pi(\cdot|s_t)) = -\log \pi(\cdot|s_t) \tag{7}$$

where $\pi$ denotes the policy, $r$ denotes the reward, $\alpha$ denotes the entropy regularization coefficient that determines the relative importance of the entropy term with respect to the reward, and $H(\pi(\cdot|s_t))$ is the entropy of the policy $\pi$ at state $s_t$.

Every action taken by the agent will generate an immediate reward $r(s_t, a_t)$, and recording the reward value can form a reward space $R(t) = \{r_1, r_2 \ldots r_t\}$. Due to the natural imbalance of network traffic, normal traffic accounts for a much larger portion of the training dataset than anomalous traffic, and the portion of different classes of anomalous traffic varies. In the design of the reward function, the corresponding reward values assigned to the agent's performance in different situations are given full consideration. The reward value design is shown in Eq. (8) as follows:

(1) When the attacker agent selects an attack category sample and the defender agent determines it as an attack and the type judgment is accurate, the attacker agent's basic reward value is set to −4 and the defender agent's basic reward value is set to +4;

(2) When the attacker agent selects an attack category sample and the defender agent determines it as an attack but misjudges the type, the attacker agent's basic reward value is set to +2 and the defender agent's basic reward value is set to +2;

(3) When the attacker agent selects an attack category sample and the defender agent determines it as normal traffic, the attacker agent's basic reward value is set to +4 and the defender agent's basic reward value is set to −4;

(4) When the attacker agent selects a normal sample and the defender agent determines it as normal traffic, the attacker agent's basic reward value is set to −1 and the defender agent's basic reward value is set to +1;

(5) When the attacker agent selects a normal sample and the defender agent determines it as an attack, the attacker agent's base reward is set to +2 and the defender agent's base reward is set to −2.

$$R_{Basic} = \begin{cases} R_a = -4, R_d = +4 & a_{a,t} \in attack, a_{d,t} \in attack, a_{d,t} = a_{a,t} \\ R_a = +2, R_d = +2 & a_{a,t} \in attack, a_{d,t} \in attack, a_{d,t} \neq a_{a,t} \\ R_a = +4, R_d = -4 & a_{a,t} \in attack, a_{d,t} \notin attack \\ R_a = -1, R_d = +1 & a_{a,t} \notin attack, a_{d,t} \notin attack \\ R_a = +2, R_d = -2 & a_{a,t} \notin attack, a_{d,t} \in attack \end{cases} \tag{8}$$

The class-weighted reward scheme (*e.g.*, +4 for detecting U2R attacks) directly addresses data imbalance in IDS, where rare attacks (0.04% of NSL-KDD training data) are often overlooked. At the same time, in order to enable the classifier to identify the minority category of abnormal traffic, this method assigns a higher reward value to the minority category of abnormal traffic, as shown in Eq. (9).

$$R = \begin{cases} 1 \cdot R_{Basic} & a_t/Total > 3\% \\ 2 \cdot R_{Basic} & 1\% < a_t/Total \leq 3\% \\ 4 \cdot R_{Basic} & a_t/Total \leq 1\% \end{cases} \tag{9}$$

### Mini-Batch data generation module

Reinforcement learning algorithms usually deal with unsupervised learning problems; however, the NSL-KDD and CSE-CIC-IDS2018 datasets used in this article are supervised and come with labels. In order to adapt the supervised dataset into the reinforcement learning framework, we adopted the Mini-Batch strategy. In "Problem Mapping", all features except the label are considered to be the current state $s_t$, the label is considered to be the current action $a_t$, and all features of the latter data except the label are considered to be the next state $s_{t+1}$. The Mini-Batch structure used is shown in Fig. 3.

The fundamental principle of the Mini-Batch strategy is predicated on two key concepts: randomness and iteration. In each training batch, Mini-Batch randomly samples a set of samples containing $[s_t, a_t, s_{t+1}]$ triplets from the original dataset to avoid repeated use of samples and ensure that the dataset is fully and evenly explored during iterative training. By using the Mini-Batch strategy in SA3C-ID, we adapt the supervised dataset to the reinforcement learning framework, enabling the intrusion detection model to be effectively trained based on the data.

### SA3C adversarial intrusion detection model

In the agent construction process described in this article, a reinforcement learning agent based on the SAC algorithm is designed to optimize the decision-making strategy by interactive learning with network traffic data. As shown in Fig. 4, the SA3C agent includes an Actor network, two V Critic networks, and two Q Critic networks.

The Actor network is responsible for the output of the agent's actions. Its input is the state $s_t$ of the current time step, and its output is the action probability distribution $\pi(a|s)$,

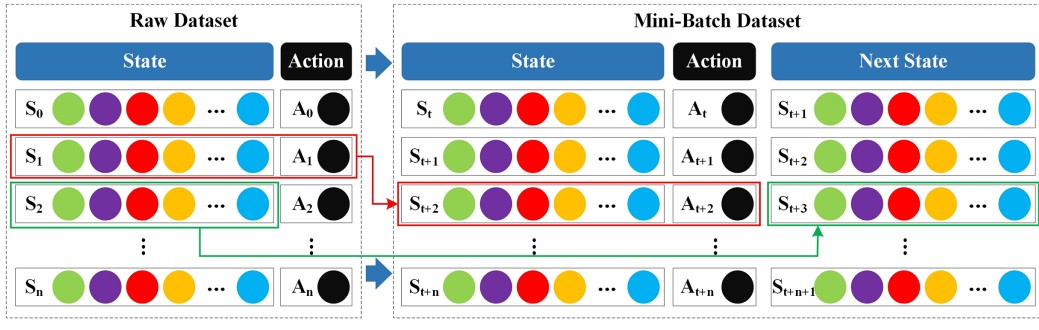

**Figure 3** Mini-Batch strategy structure.

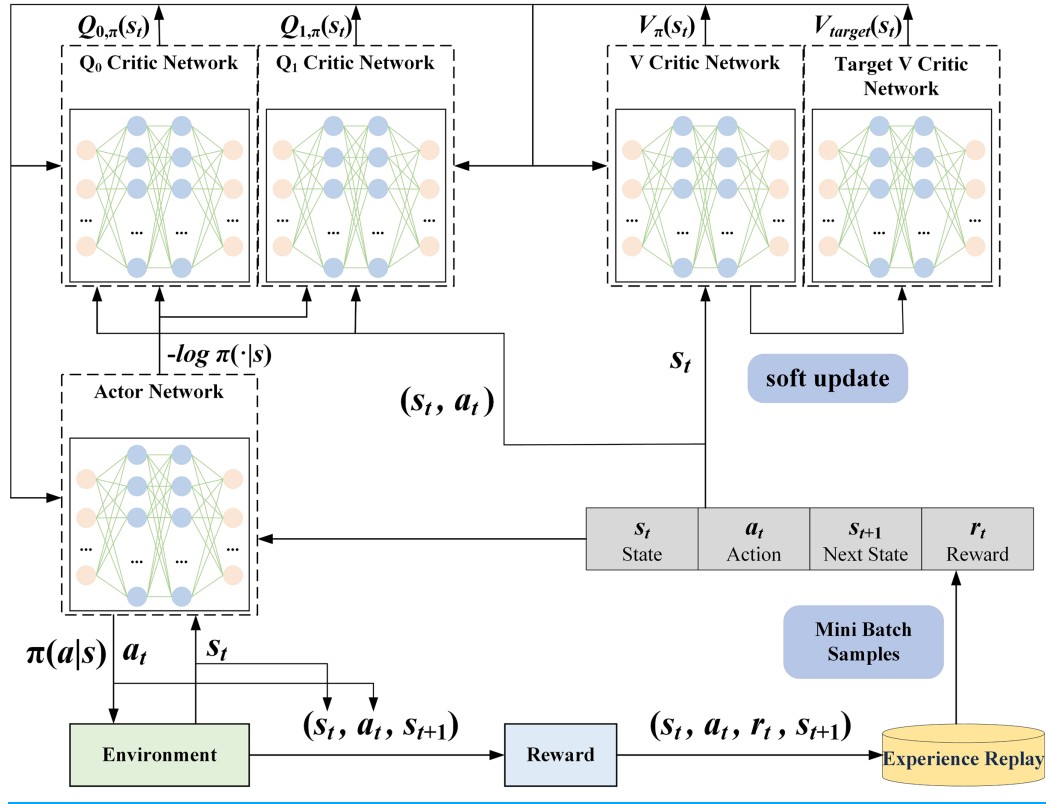

**Figure 4** SA3C agent structure.

with each action having a corresponding probability value. The agent's behavior is typically characterized by its selection of actions with higher probabilities. However, the SAC algorithm's emphasis on exploration leads the agent, at times, to opt for actions with lower probabilities to maintain the randomness and diversity of the strategy The exploration balance is usually achieved by adding an entropy term to the loss function. The Actor network loss function in this article is shown in Eq. (10).

$$L_{\text{Actor}} = -\frac{1}{|M|} \sum_{e \in M} \left( \min_{j=1,2} Q_j(s_t, a_t) - \alpha \log(\pi(a_t|s_t)) \right) \tag{10}$$

where $|M|$ denotes the number of experiences used in a training step, $Q_j(s_t, a_t)$ denotes the output value of the $j$th Q Critic network under state $s_t$ and action $a_t$. The minimum of the outputs of the two Q networks is taken to minimize overestimation. Additionally, $\alpha$ denotes the entropy regularization coefficient, which is used to encourage the Actor network to output more uncertain actions.

The V Critic network is utilized to estimate the value of a state. Its input is the state $s_t$ at the current time step, and the output is the state-value function $V(s_t)$, which represents the expected return that the agent can obtain when following the current policy in the state $s_t$. The loss function of the V Critic network is shown in Eq. (11).

$$L_{\text{V−Critic}} = \frac{1}{|M|} \sum_{e \in M} \left( Q(s_t, a_t) - E_{a'_t \sim \pi(\cdot|s_t)} \left[ \min_{j=1,2} Q_j(s_t, a'_t) - \alpha \log(\pi(a'_t|s_t)) \right] \right)^2 \quad (11)$$

where $E_{a'_t \sim \pi(\cdot|s_t)}$ represents the expectation over the action $a'_t$ sampled from the action probability distribution of the current policy $\pi$ in the state $s_t$. This article proposed the incorporation of a regularization term associated with policy entropy into the loss function of the V Critic network to enhance the model's generalization ability.

The Q Critic network is employed to evaluate the values of different actions. Its input is the state $s_t$ at the current time step, and its output is the action-state value function $Q(s_t, a_t)$, which represents the expected return that the agent can obtain by following the current policy after choosing action $a_t$ in state $s_t$. In this article, the loss function of the Q Critic network is shown in Eq. (12).

$$L_{\text{Q−Critic}} = \frac{1}{|M|} \sum_{e \in M} \left( V(s_t) - \left( r_t + \gamma E_{a'_{t+1} \sim \pi(\cdot|s_{t+1})} \left[ \min_{j=1,2} Q_j(s_{t+1}, a'_{t+1}) \right] - \alpha \log(\pi(a'_{t+1}|s_{t+1})) \right) \right)^2 \quad (12)$$

where $r_t$ represents the immediate return obtained after executing action $a_t$ in state $s_t$, while $\gamma$ is the discount factor, which controls the influence of future returns on the current value. The loss function of the Q Critic network optimizes the network parameters by minimizing the difference between the predicted value and the actual return.

As illustrated in Fig. 5, the architectures of the attacker agent and the defender agent are comprised of a shallow neural network, namely a multi-layer perceptron (MLP). The architecture of the attacker agent consists of two layers with 100 neurons in each layer, while the architecture of the defender agent comprises three layers with 150 neurons in each layer. As can be seen from Fig. 5, both the attacker agent and the defender agent take the features of the current sample as inputs, but produce different outputs depending on the differences in their respective networks. For the attacker agent in the NSL-KDD dataset context, the Actor network generates a probability distribution over 40 traffic types, encompassing normal and attack categories, *via* its policy function $\pi(a|s)$. This distribution guides the selection of a specific traffic type, after which a corresponding real sample is extracted from the dataset.

The attacker's selection process is driven by the reward function defined in "Problem Mapping", which assigns higher rewards for successfully evading the defender (*e.g.*, when an attack is misclassified as normal traffic). Through iterative adversarial training, the attacker agent optimizes its policy to maximize cumulative rewards, prioritizing attack

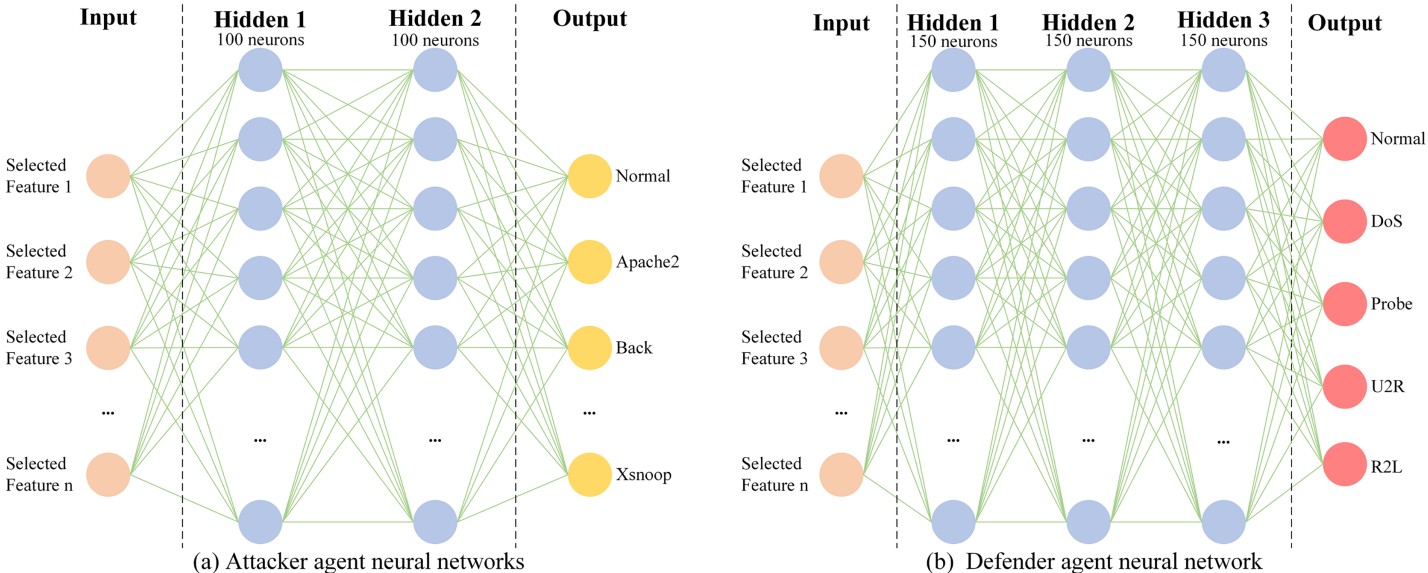

**Figure 5 Neural network structure of the attacker agent and the defender agent.** (A) Attacker agent neural networks (B) Defender agent neural network.

categories that challenge the defender's current detection strategy. This adaptive mechanism allows the attacker to learn context-aware selection patterns, such as focusing on minority-class attacks.

For the defender agent, the output is a categorical prediction of the intrusion category, enabled by its three-layer MLP architecture. The defender's policy is updated in parallel, leveraging the prioritized experience replay buffer to learn from high-impact interactions with the attacker. Together, the attacker's strategic sample selection and the defender's adaptive classification form a closed-loop adversarial system, where both agents refine their strategies through reinforcement learning to simulate realistic attack-defense dynamics.

### Priority experience replay

SA3C-ID improves the training speed and stability of the algorithm by introducing the prioritized experience replay mechanism into the SAC algorithm. In the network intrusion detection of this article, the state at time $t$ is $s_t$, the action $a_t$ is selected based on the probability distribution policy $\pi(a|s_t)$ output by the Actor network. The action $a_t$ is input into the state $s_t$ to obtain the reward $r_t$ at the current time step, the next state $s_{t+1}$, and a boolean value $d_t$ indicating whether the interaction between the agent and the environment has reached the terminal state. Meanwhile, each piece of experience is associated with a weight parameter $w_t$ calculated by the temporal-difference error (TD error). The TD error calculation formula is shown in Eq. (13).

$$\delta_t = Q(s_t, a_t) - (r_t + \gamma(1 - d_t) \cdot V(s_{t+1})) \tag{13}$$

where $\delta_t$ represents the TD Error of the current time step, $Q(s_t, a_t)$ represents the expected return of taking action $a_t$ in state $s_t$, $\gamma$ represents the discount factor, and $V(s_{t+1})$

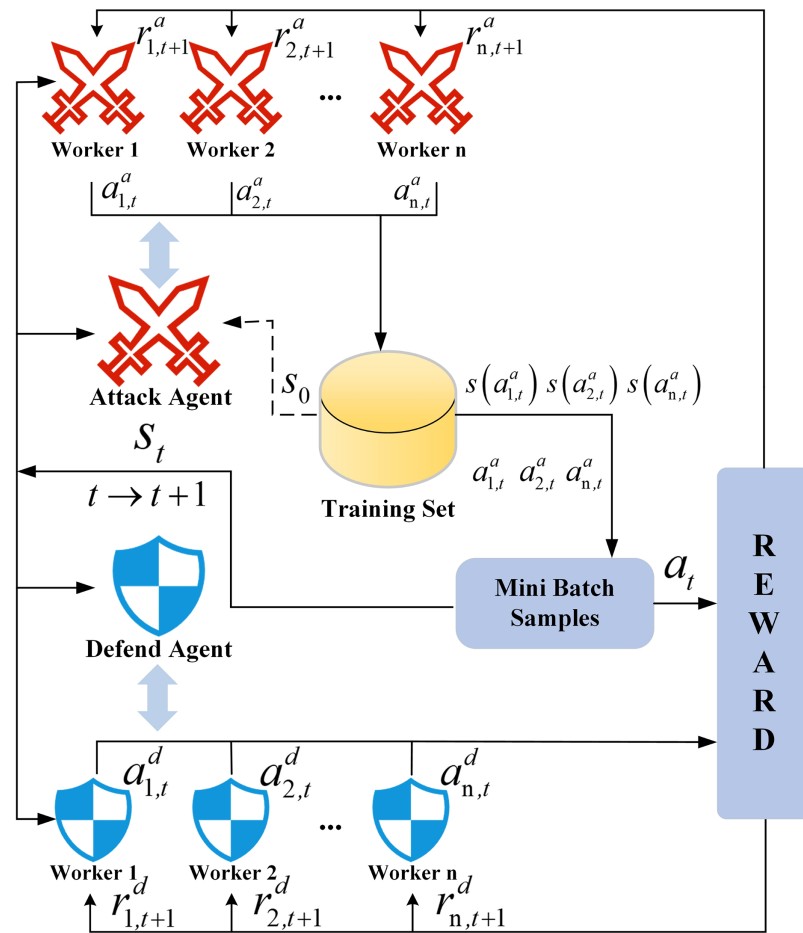

**Figure 6** SA3C algorithm asynchronous training architecture.

represents the expected return of the next state $s_{t+1}$. Each piece of experience is stored in the experience replay buffer in the form of a six-tuple $<s_t, a_t, r_t, s_{t+1}, d_t, w_t>$. By assigning different priority weights to different experiences, the temporal correlation between samples is broken, and the utilization efficiency of samples is improved. During the agent training process, samples from the replay buffer are randomly selected according to their priorities to update the parameters of the neural network, aiming to enhance the stability and efficiency of learning.

### Asynchronous training strategy

The asynchronous training mechanism is designed to tackle dynamic attack-defense dynamics and convergence delays in centralized RL frameworks. During the training process of the SA3C model, multiple worker threads are created. Each thread contains an independent attacker agent, a defender agent, and a shared environment copy. All worker threads execute training tasks concurrently. That is to say, each thread independently interacts with the environment, collects data, and updates the parameters of the policy

**Algorithm 1    Soft adversarial asynchronous actor-critic intrusion detection (SA3C-ID).**

**Input:** Train Dataset $D_{Train}$, Test Dataset $D_{Test}$, Number of workers $N_{workers}$

**Output:** Detection results of policy $\pi_\theta$ on test dataset $D_{Test}$

01: Preprocess train and test datasets

02: Initialize datasets by SABPIO algorithm: $D'_{Train} \leftarrow D_{Train}$, $D'_{Test} \leftarrow D_{Test}$

03: Initialize global experience replay pool: $B^a_{globle} \leftarrow 0$, $B^d_{globle} \leftarrow 0$

04: Initialize $\pi_\theta$ (defender policy) and $\pi_\varphi$ (attacker policy) randomly

05: Initialize shared parameters $\theta$ and $\varphi$ for $\pi_\theta$ and $\pi_\varphi$ respectively

06: Initialize worker threads based on $N_{workers}$

07: **foreach** worker **in** worker threads **to do**

08:      **for** epoch = 1 **to** epochs **do**

09:          Initialize the state $s_0 \leftarrow$ random sampling from $D'_{Train}$

10:          Choose a set of actions $A_{a,t}$ based on $\pi_\varphi(s_0)$

11:          Initialize local Mini-Batch buffer $M \leftarrow$ random sampling from $D'_{Train}$, and labels all belong to $A_{a,t}$

12:              **while not** done **do**

13:                  Choose action $a^i_{d,t}$ based on $\pi_\theta(s_t)$ for each state $s^i_t$ in $M$

14:                  Obtain rewards for the attacker and the defender: $r_{a,t+1}$, $r_{d,t+1}$

15:                  Choose new set of actions $A_{a,t+1}$ based on $\pi_\varphi(s_t)$

16:                  Update local Mini-Batch buffer $M \leftarrow$ random sampling from $D_{Train}$, and labels all belong to $A_{a,t+1}$

17:                  Calculate the priority weight $w_t$ of each attacker experience $\langle s_t, a_{a,t}, r_{a,t+1}, s_{a,t+1} \rangle$ and each defender experience $\langle s_t, a_{d,t}, r_{d,t+1}, s_{d,t+1} \rangle$

18:                  Store attacker experience $B^a_{globle} \leftarrow \langle s_t, a_{a,t}, r_{a,t+1}, s_{a,t+1}, d_t, w_t \rangle$

19:                  Store defender experience $B^d_{globle} \leftarrow \langle s_t, a_{d,t}, r_{d,t+1}, s_{d,t+1}, d_t, w_t \rangle$

20:                  Update worker attacker agent by sampling $M$ experience from $B^a_{globle}$ based on their weights $w_t$

21:                  Update worker defender agent by sampling $M$ experience from $B^d_{globle}$ based on their weights $w_t$

22:              **end while**

23:              Update global attacker agent by worker attacker agent

24:              Update global defender agent by worker defender agent

25:      **end for**

26: **end foreach**

27: Test policy $\pi_\theta$ on test dataset $D'_{Test}$

28: **Return** Detection results

network and value network of its respective agent. Subsequent to each update, the agent parameters in the worker threads are not immediately synchronized to the main network. Instead, the updated parameters are asynchronously transferred to the main network at an opportune time.

In the asynchronous update method, each worker thread is capable of exploring different parts of the environment independently, thus avoiding the issue of getting trapped in local optima. In this article, the aforementioned approach is used to achieve efficient synchronization of the parameters of the attacker agent and the defender agent, ensuring the stability and effectiveness of the entire training process. Meanwhile, the computing resources of multi-core CPUs are fully utilized to accelerate the training process.

The architecture of the SA3C algorithm is shown in Fig. 6. The integration of the adversarial training mechanisms of the attacker and defender agents enhances the model's detection accuracy and adaptability in complex network environments. The adoption of an asynchronous update strategy maximizes the utilization of computing resources, significantly reduces the training cycle, and provides scientific decision-making support for real-time network protection.

SA3C-ID model algorithm is shown in Algorithm 1. First, methods such as one-hot encoding and standardization are used to integrate, clean, transform, and standardize the dataset. Second, the SABPIO algorithm is employed to simulate the pigeon flock migration behavior and search for the optimal feature subset in the feature space, eliminating redundant features to reduce the data dimension. Third, the Mini-Batch training strategy is utilized to re-encode the data after feature selection. Then, the re-encoded data is input into the SA3C adversarial intrusion detection module. The pre-processed data is converted into states, actions, and reward values. The prioritized experience replay buffer technology is used to stably and efficiently optimize the network parameters, and the computing resources are fully utilized through asynchronous updates. Finally, an adversarial training mechanism is introduced, where the attacker and defender agents are set up for iterative confrontation. The agents on both sides continually refine their classification strategies through the use of reinforcement learning, ultimately achieving effective detection of intrusion traffic.

# EXPERIMENTAL RESULTS AND DISCUSSION

## Experimental dataset

(1) NSL-KDD dataset

The NSL-KDD dataset is a revised version of the famous KDD99 dataset. Compared with KDDCUP99, the NSL-KDD dataset has been optimized in terms of the number of instances, which effectively reduces the impact of data randomness. The NSL-KDD dataset contains 148,517 records, each of which contains 41 feature attributes and one class identifier, including 39 attack categories. The training set has 125,973 data points,

**Table 3 The number of categories in the NSL-KDD dataset.**

| Sample type | Number of training sets | Number of test sets |
|---|---|---|
| Normal | 67,341 | 9,711 |
| Probe | 11,656 | 7,456 |
| DoS | 45,927 | 2,421 |
| U2R | 114 | 1,436 |
| R2L | 934 | 1,520 |

including 22 types of attacks, and the test set has 22,544 data points, including another 17 types of attacks. The distribution of each category in the dataset is shown in Table 3.

(2) CSE-CIC-IDS2018 dataset

The CSE-CIC-IDS2018 dataset is a collection of network traffic data that was gathered by the Canadian Institute for Cybersecurity (CIC). The CSE-CIC-IDS2018 dataset was constructed based on real-world network traffic captures and encompasses a wide range of cyber intrusions and normal activities. Since there is no unified test set and training set in CSE-CIC-IDS2018, in this article, to ensure the validity and reliability of our experimental results, we employed a stratified sampling approach to partition the merged dataset. Specifically, we allocated 70% of the data for training and 30% for testing, maintaining the original class distribution within both subsets. The number of each category in the dataset is shown in Table 4.

## Evaluation indicators

In network intrusion detection tasks, evaluation indicators based on confusion matrices are usually used. When a network attack is correctly predicted, it is a true positive (TP); when normal traffic is correctly predicted, it is a true negative (TN); when normal traffic is incorrectly predicted, it is a false positive (FP); when a network attack is incorrectly predicted, it is a false negative (FN). The confusion matrix is shown in Table 5.

Based on the above confusion matrix, the evaluation metrics in this article include accuracy (Acc), detection rate (DR), precision (Pre) and F1-score as follows:

Accuracy refers to the proportion of samples correctly predicted by the model, which indicates the overall prediction accuracy of the model. The calculation method is shown in Eq. (14).

$$Acc = \frac{TP + TN}{TP + TN + FP + FN}. \tag{14}$$

Detection rate indicates the ability to correctly identify true positive samples and the proportion of intrusion events successfully detected by the model. The calculation method is shown in Eq. (15).

$$DR = \frac{TP}{TP + FN}. \tag{15}$$

**Table 4 The number of categories in the CSE-CIC-IDS2018 dataset.**

| Sample type | Number of training sets | Number of test sets |
|---|---|---|
| Benign | 3,909,463 | 2,022,695 |
| Bot | 239,397 | 42,913 |
| DDOS attack-LOIC-UDP | 1,211 | 519 |
| DDOS attack-HOIC | 479,394 | 189,067 |
| DoS attacks-SlowHTTPTest | 13,624 | 5,838 |
| DoS attacks-GoldenEye | 29,019 | 12,436 |
| DoS attacks-Hulk | 358,077 | 76,796 |
| DoS attacks-Slowloris | 7,200 | 3,085 |
| SSH-Bruteforce | 71,969 | 45,354 |
| FTP-BruteForce | 27,542 | 11,803 |
| Infilteration | 136,309 | 24,295 |
| Brute Force-Web | 528 | 83 |
| Brute Force-XSS | 199 | 31 |
| SQL Injection | 75 | 12 |

**Table 5 Confusion matrix.**

| | | Predicted | |
|---|---|---|---|
| | | Positive | Negative |
| Actual | Positive | True positive (TP) | False negative (FN) |
| | Negative | False positive (FP) | True negative (TN) |

Precision refers to the proportion of positive samples correctly identified by the model. In network intrusion detection, it refers to the accuracy of the model among all samples identified as intrusions. The calculation method is shown in Eq. (16).

$$Pre = \frac{TP}{TP + FP}. \tag{16}$$

F1-score is the harmonic mean of precision and detection rate, which combines the accuracy and completeness of the model and helps evaluate the balance between detection rate and precision of the model. The calculation method is shown in Eq. (17). In the context of intrusion detection systems (IDS), where class imbalance (*e.g.*, normal traffic vastly outnumbering rare attacks) is a critical challenge, the F1-score serves as a pivotal metric. Unlike simple accuracy, it balances precision and detection rate to evaluate performance on minority attack classes.

$$\text{F1-score} = 2 * \frac{Pre * DR}{Pre + DR} = \frac{2 * TP}{2 * TP + FP + FN}. \tag{17}$$

## Experimental results

This article conducted the evaluation experiments of the proposed method on a 64-bit Windows 11 operating system. The device hardware includes 32 GB RAM, Intel(R) Core

**Table 6 SA3C-ID model training parameters.**

| Parameters | Value |
|---|---|
| Number of pigeons $NUM_{pigeon}$ | 128 |
| Number of iterations $NUM_t$ | 100 |
| Number of annealing iterations $NUM_{at}$ | 100 |
| Weighting factor of Pearson $\omega_1$ | 0.4 |
| Weighting factor of Cosine $\omega_2$ | 0.4 |
| Weighting factor of Jaccard $\omega_3$ | 0.2 |
| Weighting factor of number of selections $\lambda$ | 0.0075 |
| Weighting factor of attenuation factor $\beta$ | 0.8 |
| Learning rate of SA3C-ID | 0.01 |
| Mini-Batch buffer size $|M|$ | 200 |
| Discount factor $\gamma$ | 0.001 |
| Entropy regularization coefficient $\alpha$ | 0.2 |
| Number of epoch | 100 |
| Number of worker | 4 |
| Experience replay pool size | 1,500 |

**Table 7 Ablation experimental results.**

| Model | Acc | Pre | DR | F1-score | Train time (sec) |
|---|---|---|---|---|---|
| Baseline | 0.8415 | 0.8427 | 0.8415 | 0.8397 | 9,099 |
| + SABPIO | 0.8864 | 0.9573 | 0.8427 | 0.8401 | 4,253 |
| + Asynchronous | 0.9022 | 0.9656 | 0.8631 | 0.8993 | 7,084 |
| SA3C-ID | 0.9192 | 0.9741 | 0.8821 | 0.9258 | 3,817 |

(TM) i5-14600KF processor and NVIDIA GeForce RTX 4070 Ti SUPER graphics card. In the experimental environment, we used Tensorflow 2.10.0 in python 3.8.12 for model development, and used library functions such as Scikit-Learn, NumPy and Pandas for data processing. The model parameters of SA3C-ID are shown in Table 6.

### Ablation experiments

In order to verify the effectiveness of each module of SA3C-ID model, we conducted ablation experiments based on the NSL-KDD dataset for four models: baseline SAC model, SABPIO +basic SAC model, asynchronous training + basic SAC model, and SA3C-ID model. The experimental results are shown in Table 7.

The ablation experiment based on the NSL-KDD dataset, as shown in Fig. 7, aimed to assess the influence of various components on model performance. The baseline model had an Acc of 0.8415, Pre of 0.8427, DR of 0.8415, F1-score of 0.8397, and a training time of 9,099 s. When the SABPIO module was added to the baseline, the Acc increased to 0.8864, Pre to 0.9573, DR slightly rose to 0.8427, F1-score became 0.8401, and the training time decreased significantly to 4,253 s, indicating that while it improved some metrics and reduced training time, the enhancement in DR and F1-score was limited. Incorporating

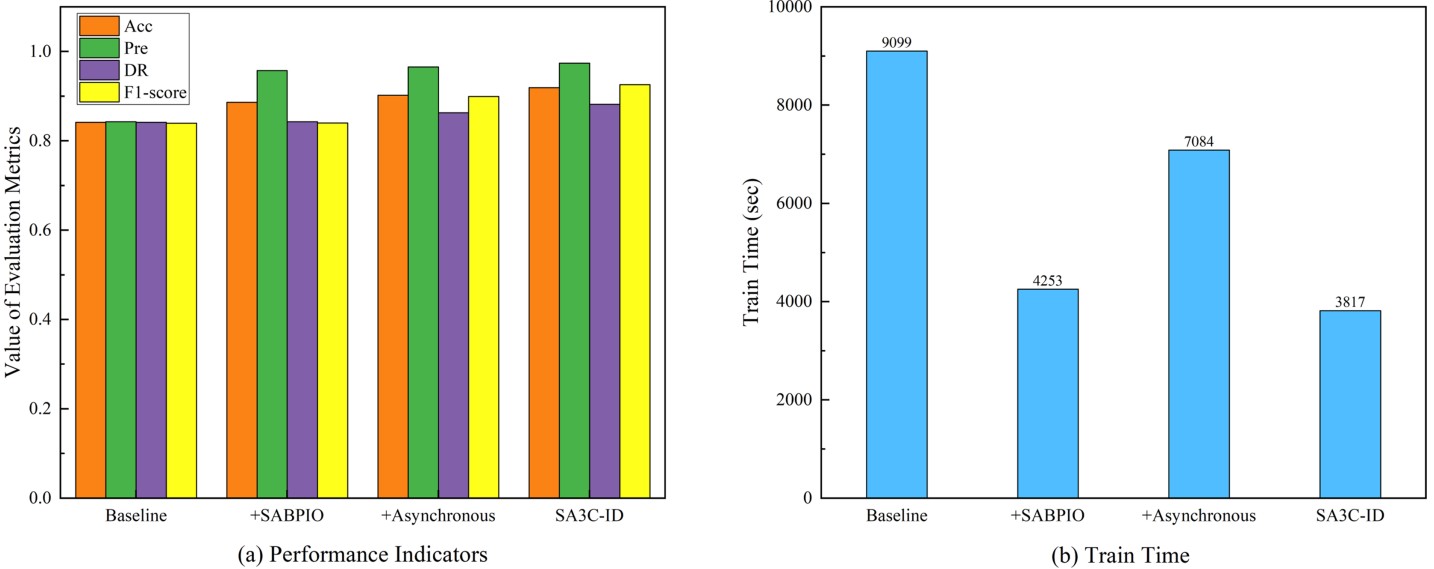

**Figure 7** **Ablation experimental results of SA3C-ID model.** (A) Performance indicators. (B) Training time.

asynchronous training led to an Acc of 0.9022, Pre of 0.9656, DR of 0.8631, F1-score of 0.8993, and a training time of 7,084 s, which was shorter than the baseline yet longer than with the SABPIO module alone, signifying an improvement in detection ability and performance balance. The SA3C-ID model, combining the baseline SAC model with other optimization modules, achieved the optimal performance with an Acc of 0.9192, Pre of 0.9741, DR of 0.8821, F1-score of 0.9258, and a training time of 3,817 s, thus fully demonstrating the effectiveness of this integrated optimization approach.

Although the SABPIO module did not improve model performance when used alone, its reduced training time suggests its potential in accelerating the training process. Asynchronous training performs well in improving model performance and accelerating the training process. SA3C-ID model achieved significant improvements in diagnosis rate, F1-score and training time compared to the baseline model.

### SA3C-ID performance evaluation

This study used the training and test sets of the NSL-KDD dataset to verify the effectiveness of the proposed detection method, and comprehensively evaluated the performance through accuracy, precision, detection rate, and F1-score.

Figures 8A and 8B respectively show the changing trends of the training reward value and training loss of SA3C-ID model on the NSL-KDD dataset. From the experimental results in Fig. 8, it can be observed that as the number of training iterations increases, the training loss of SA3C-ID attack and defense agents continues to decrease and tends to converge after about 80 iterations.

Figures 9A and 9B show the confusion matrix of SA3C-ID model for attack classification in the NSL-KDD dataset. The horizontal axis is the predicted data and the vertical axis is the real data. The classification effect of each type of label is distinguished by

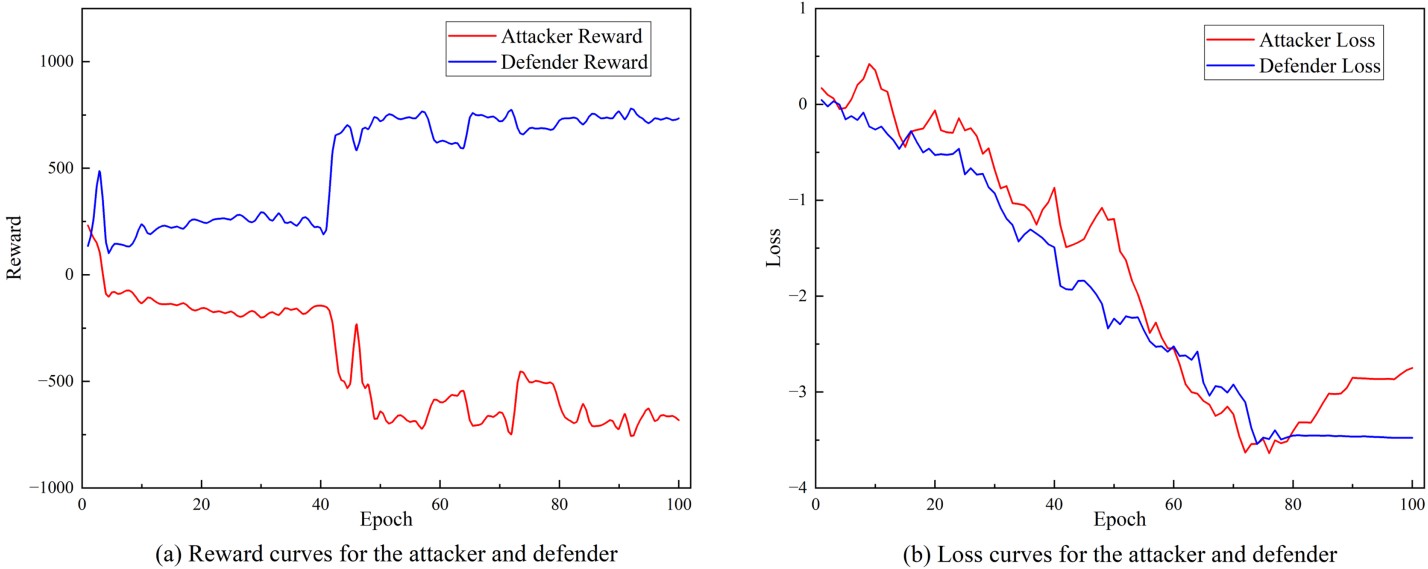

**Figure 8** **Training curve of SA3C-ID on NSL-KDD dataset.** (A) Reward curves for the attacker and defender. (B) Loss curves for the attacker and defender.

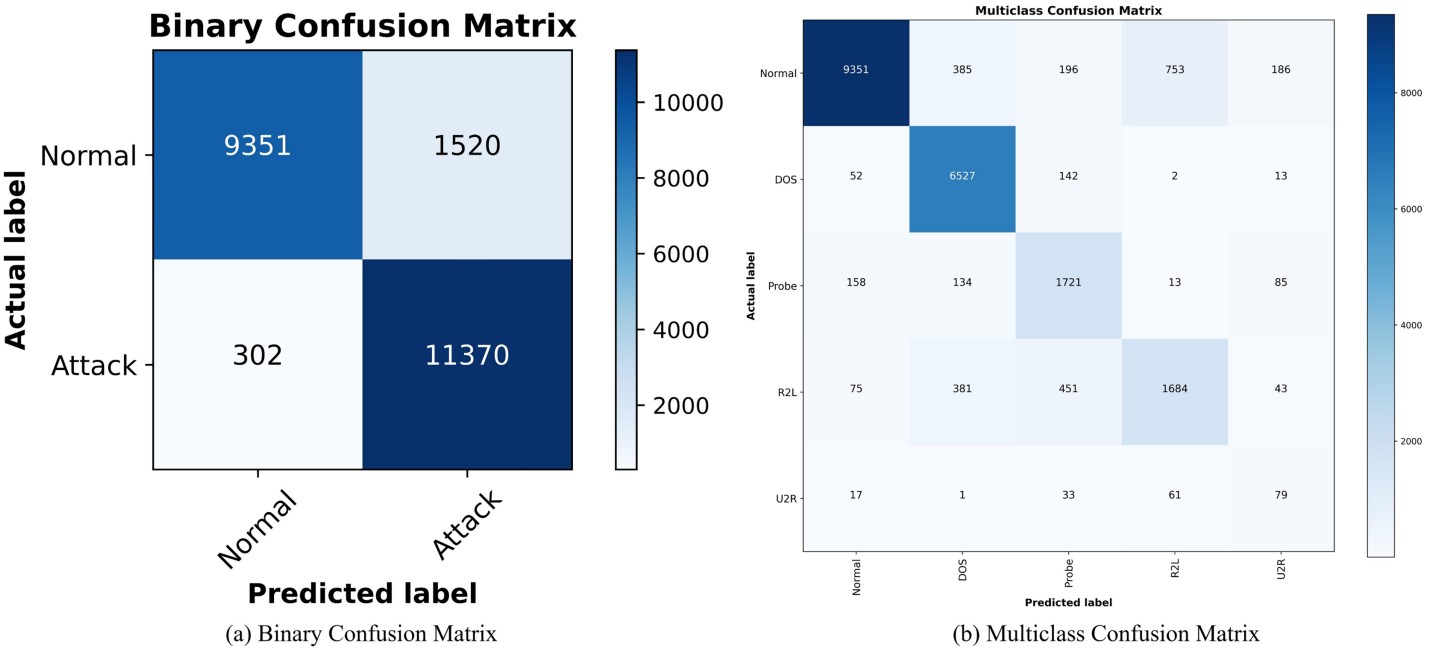

**Figure 9** **Confusion matrix of SA3C-ID model based on NSL-KDD dataset.** (A) Binary confusion matrix. (B) Multiclass confusion matrix.

the depth of color in the same row. The intrusion detection capability of SA3C-ID can be intuitively observed through the confusion matrix. For normal traffic, SA3C-ID misclassifies only 1.2% of samples as attacks, demonstrating its capability to reduce false alarms, which is a persistent challenge in IDS.

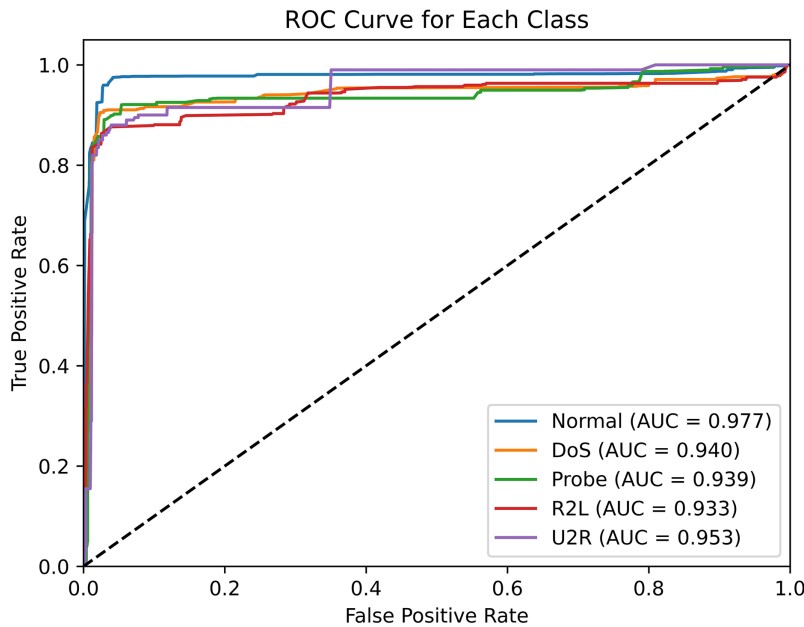

**Figure 10 ROC curve of SA3C-ID model in NSL-KDD dataset.**

The ROC curve is a graph that illustrates the relationship between TPR and FPR, with a larger area under the curve (AUC) value indicating better performance. As illustrated in Fig. 10, the ROC curve of SA3C-ID model in the NSL-KDD dataset demonstrates that the AUC values of different categories are distinct. Specifically, the Normal class achieved an AUC of 0.977, reflecting strong recognition of normal traffic. For minority attack classes, the User-to-Root (U2R) class, which contains only 54 samples in the training set, achieved an AUC of 0.953; the Remote-to-Local (R2L) class, with 1,126 training samples, achieved an AUC of 0.968. While these values indicate relatively high detection potential for minority classes, the U2R class's AUC is 2.4% lower than the Normal class, suggesting room for improvement in detecting extremely rare attacks. This gap highlights the challenge of imbalanced data, where models may underperform on classes with insufficient training examples.

*Comparative experiment*

In this section, we compare SA3C-ID model with other intrusion detection models on the NSL-KDD and CSE-CIC-IDS2018 datasets, including Logistic regression, RF, CNN, LSTM, DQN, DDQN (*Lopez-Martin, Carro & Sanchez-Esguevillas, 2020*), AE-RL (*Caminero, Lopez-Martin & Carro, 2019*), A3C (*Grondman et al., 2012*), AE-SAC (*Li et al., 2023*), HCRNN (*Khan, 2021*) and other models. It should be noted that since most models do not give their full training results on the CSE-CIC-IDS2018 dataset, we only compared the accuracy and F1-score in the CSE-CIC-IDS2018 dataset.

Figure 11 shows the comparison results of the SA3C-ID model with other models in the NSL-KDD dataset. The experimental results show that compared with the other nine methods, SA3C-ID model achieves the highest indicators, with accuracy, precision,

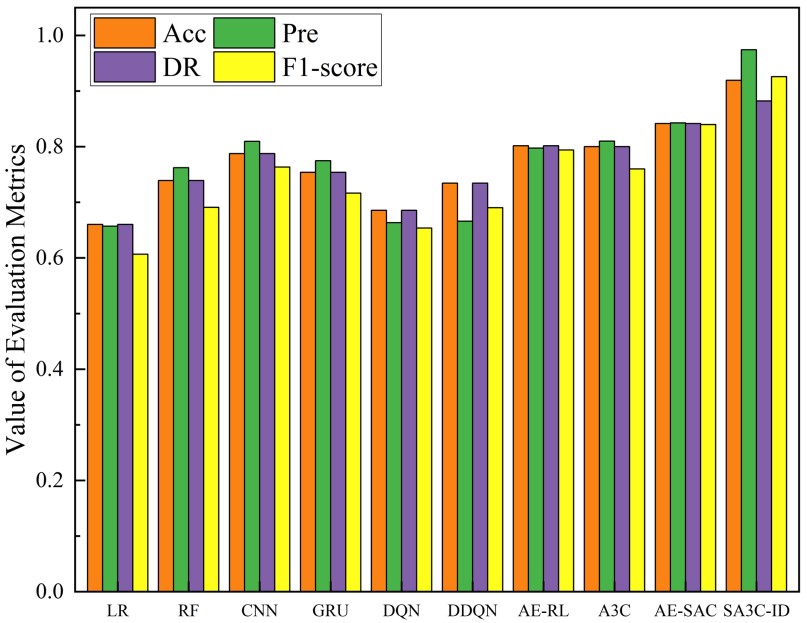

**Figure 11** **Performance of SA3C-ID model and other models in the NSL-KDD dataset.**

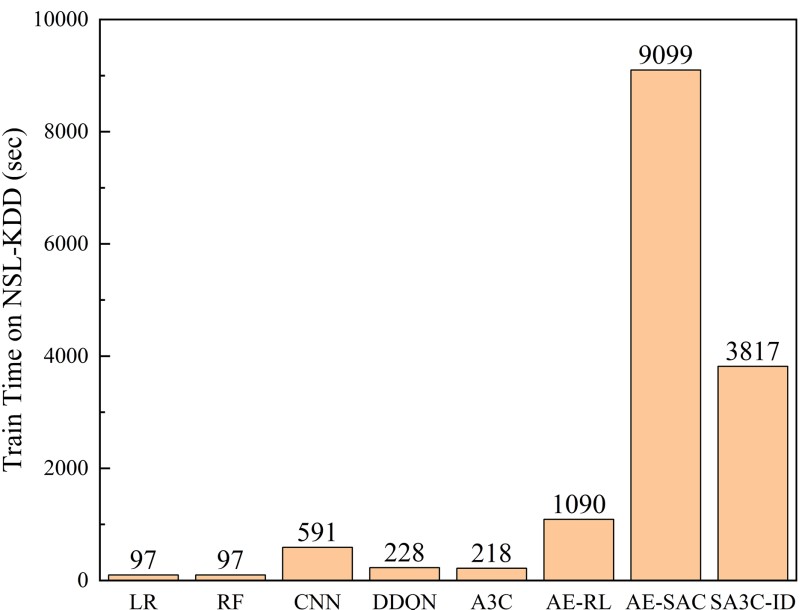

**Figure 12** **Training time between SA3C-ID model and other models in NSL-KDD dataset.**

detection rate, and F1-score reaching 0.9192, 0.9741, 0.8821, and 0.9258 respectively. The F1-score is improved by about 16.6% and 10.3% compared with AE-RL and AE-SAC respectively. SA3C-ID model starts from the problem of sample class imbalance in the intrusion detection dataset, solves the problem of low detection performance of most

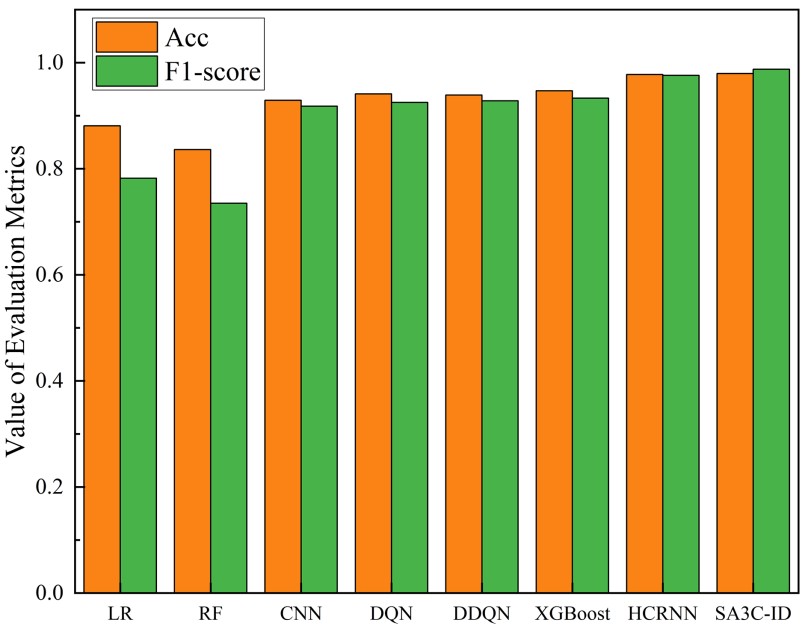

**Figure 13 Performance of SA3C-ID model and other models in CSE-CIC-IDS2018 dataset.**

intrusion detection models for minority class samples, and solves the problem of feature redundancy based on the data itself, thus achieving the best results in the comparative experiment. The detection rate of SA3C-ID in Fig. 11 is lower than that of other indicators because the model reduces the degree of attention to normal samples to a certain extent in order to enhance the detection ability of minority class attack samples.

Figure 12 shows the comparison of the training time of the SA3C-ID model with other models in the NSL-KDD dataset. The results show that compared with general machine learning, deep learning, and reinforcement learning algorithms, the adversarial training algorithms AE-RL, AE-SAC, and SA3C-ID require longer training time. However, since network intrusion detection is trained offline, in actual application scenarios, as long as the training is completed before the model is deployed, it will not interfere with real-time network intrusion detection. At the same time, in practical applications, SA3C-ID only needs to deploy the Actor network, and the network parameters have been determined in the training phase. Therefore, the operating efficiency in practical applications is not affected by the training time. The prediction time of the SA3C-ID model in the NSL-KDD and CSE-CIC-IDS2018 data sets is as follows. In the NSL-KDD test set, with a sample size of 22,543, the prediction time is 0.36 s, which equates to a prediction time of 15.97 microseconds for a single sample. Notably, SA3C-ID's single-sample inference time (15.97 μs) significantly outperforms traditional RL models (DQN: 58 μs, A3C: 42 μs), meeting the real-time detection requirements of IoT/5G devices (industry standard ≤20 μs). This efficiency stems from SABPIO's dimensionality reduction and asynchronous parameter updates.

As illustrated in Fig. 13, the SA3C-ID model demonstrates superior performance in comparison to other models within the CSE-CIC-IDS2018 dataset. Compared with the other 7 methods, the SA3C-ID model has the highest accuracy and F1-score. Since most models do not give their full training results on the CSE-CIC-IDS2018 dataset, we only compare the accuracy and F1-score here. In the CSE-CIC-IDS2018 dataset, the accuracy, precision, detection rate, and F1-score of SA3C-ID reached 0.9796, 0.9953, 0.9799, and 0.9876 respectively, and the F1-score was improved by about 5.85% and 1.19% compared with XGBoost and HCRNN, respectively.

## Discussion

In terms of the experimental results, although SA3C-ID model demonstrates high detection performance on the NSL-KDD and CSE-CIC-IDS2018 datasets, it still has certain limitations.

The findings of the ablation experiments indicate that while SA3C-ID model demonstrates commendable overall performance, the SABPIO module exhibits no substantial enhancement in terms of DR and F1-score when utilized in isolation. Integrating the SABPIO feature selection module into this baseline reduced training time by 53% (from 9,099 to 4,253 s) but yielded minimal performance gains, indicating that dimensionality reduction alone cannot resolve class imbalance challenges without strategic adversarial interaction.

Notably, the significant gains in the asynchronous training variant highlight that parallelized agent interactions are critical for effective adversarial learning. By enabling independent exploration across worker threads, asynchronous training increases the frequency of minority-class sample encounters, as reflected in the 14.5% higher recall for rare attack categories compared to the baseline. When combined with SABPIO, the model achieves a 16.6% improvement in F1-score over the state-of-the-art AE-RL, demonstrating that dimensionality reduction and adaptive training strategies act in concert to enhance both speed and accuracy. During asynchronous training, SABPIO reduces the feature space from 41 to 6 dimensions in NSL-KDD, thereby decreasing state space complexity by 85%. The low-dimensional features reduce the computational load per state (such as lowering the input dimension of neural networks), enabling each worker thread to process more distinct states within the same time frame. As demonstrated in the ablation tests (Table 7), this synergy minimizes redundant computations.

Furthermore, the model's detection performance varies across different attack categories, highlighting its advantage in handling class imbalance. In the NSL-KDD dataset, U2R attack samples are scarce, accounting for only 0.04% of the training set. Yet, the SA3C-ID model attains an AUC value of 0.953 for U2R. This benefit stems from the class-weighted adaptive reward function, which gives +4 for correctly detecting U2R attacks (4 times the reward for normal traffic). However, compared to the detection of the Normal category, there's room for improvement, implying that the model's strategies for small-sample attacks can be optimized. Subsequent research should focus on enhancing the model's structure and training mechanism to boost its detection of various attacks, improving stability across different scenarios.

## CONCLUSION

In an era of increasingly complex network architectures and sophisticated cyber threats, network security has grown critically important. Traditional intrusion detection methods face challenges like inadequate feature extraction, high model complexity, and data imbalance, leading to low detection efficiency and difficulties in identifying complex attacks. To address these challenges, this article presents SA3C-ID, an adversarial intrusion detection model based on reinforcement learning. Its core innovations include the integration of the SABPIO feature selection module, an adaptive reward function, and asynchronous parameter updates. SABPIO streamlines training by eliminating redundant high-dimensional features, the adaptive reward function prioritizes minority-class attacks, and asynchronous updates enable real-time strategy optimization against evolving threats. This design ensures the model detects rare intrusions while handling majority-class normal traffic, filling a critical gap in existing models. Ablation and comparative experiments on NSL-KDD and CSE-CIC-IDS2018 datasets show SA3C-ID outperforms other models, achieving F1-scores of 92.58% and 98.76% respectively, with a single-sample inference time of 15.97 microseconds-meeting low-latency needs in 5G and IoT environments where traditional RL models often exceed 50 microseconds in delay and fail real-time monitoring.

While SA3C-ID demonstrates strong performance, opportunities for improvement exist. Future work will focus on leveraging generative adversarial network (GAN) to generate realistic attack samples and alleviate data imbalance by enhancing minority-class representation. Additionally, optimizing the SABPIO algorithm, refining model architecture and training mechanisms, and improving adaptability to dynamic network attacks will further boost performance.

### Funding

This work was supported by the Major Science and Technology Programs in Henan Province (No. 241100210100), the Henan Provincial Science and Technology Research Project (No. 252102211085, No. 252102211105), the Special Project for Research and Development in Key areas of Guangdong Province (No. 2021ZDZX1098), the China Higher Education Institution Industry-University-Research Innovation Fund (No. 2021FNB3001, No. 2022IT020), the Stabilization Support Program of Science, Technology and Innovation Commission of Shenzhen Municipality (No. 20231128083944001) and Henan Provincial Key Scientific Research Project Program of Higher Education Institutions (No. 24A520042).There was no additional external funding received for this study. The funders had no role in study design, data collection and analysis, decision to publish, or preparation of the manuscript.

## Grant Disclosures

The following grant information was disclosed by the authors:

Major Science and Technology Programs in Henan Province: 241100210100.

Henan Provincial Science and Technology Research Project: 252102211085 and 252102211105.

Special Project for Research and Development in Key Areas of Guangdong Province: 2021ZDZX1098.

China Higher Education Institution Industry-University-Research Innovation Fund: 2021FNB3001 and 2022IT020.

Stabilization Support Program of Science, Technology and Innovation Commission of Shenzhen Municipality: 20231128083944001.

Henan Provincial Key Scientific Research Project Program of Higher Education Institutions: 24A520042.

## Competing Interests

Lei Wang is employed by China Research Institute of Radio Wave Propagation. Songze Li is employed by Henan Xinda Wangyu Technology Co. Ltd.

## Author Contributions

- Wanwei Huang conceived and designed the experiments, analyzed the data, authored or reviewed drafts of the article, and approved the final draft.
- Haobin Tian conceived and designed the experiments, performed the experiments, analyzed the data, performed the computation work, prepared figures and/or tables, authored or reviewed drafts of the article, and approved the final draft.
- Lei Wang performed the experiments, analyzed the data, authored or reviewed drafts of the article, and approved the final draft.
- Sunan Wang performed the experiments, authored or reviewed drafts of the article, and approved the final draft.
- Kun Wang performed the experiments, authored or reviewed drafts of the article, and approved the final draft.
- Songze Li analyzed the data, authored or reviewed drafts of the article, and approved the final draft.

## Data Availability

The source code and raw data is available in the Supplemental File.

The original NSL-KDD dataset is available at: https://web.archive.org/web/20150205070216/http://nsl.cs.unb.ca/NSL-KDD. It was originally available at (no longer working): https://www.unb.ca/cic/datasets/nsl.html.

The CSE-CIC-IDS2018 dataset is available at: https://www.unb.ca/cic/datasets/ids-2018.html.

## Supplemental Information

Supplemental information for this article can be found online at http://dx.doi.org/10.7717/peerj-cs.3089#supplemental-information.

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
