# Peer review of "SA3C-ID: a novel network intrusion detection model using feature selection and adversarial training"

_PeerJ Computer Science, doi:10.7717/peerj-cs.3089_

## Round 0.1 · original submission · Major Revisions

Reviewers have now commented on your paper. The reviewers have raised concerns regarding the experimental setup and experimental results, as well as methodology and discussion. These issues require a major revision. Please refer to the reviewers’ comments at the end of this letter; you will see that they advise you to revise your manuscript. If you are prepared to undertake the work required, I would be pleased to reconsider my decision. Please submit a list of changes or a rebuttal against each concern when you submit your revised manuscript.

Thank you for considering PeerJ Computer Science for the publication of your research.

With kind regards,

Reviewer 1 ·

Basic reporting

The writing is generally clear and professional, with no obvious spelling or grammar errors. However, minor improvements in language flow and connection could enhance readability. The literature review is extensive and well-articulated, providing a solid background and context for the research. The article is structured professionally, with well-made figures that effectively support the text. All relevant results are presented in a self-contained manner, aligning with the hypotheses. It is recommended that the authors include the full names of abbreviations, such as "PIO algorithm" in line 30 of the abstract and "SABPIO" in line 262, to ensure clarity for all readers.

Experimental design

The research presented is original and falls within the Aims and Scope of the journal. The research question is well-defined, relevant, and meaningful, addressing an identified knowledge gap in the field. The investigation is rigorous and adheres to high technical and ethical standards. The methods are described in sufficient detail, allowing for replication of the study, which is crucial for validating the findings.

Validity of the findings

The findings of the study are promising and contribute valuable insights to the field. The results are robust, statistically sound, and well-controlled, providing a strong foundation for the conclusions drawn. The authors have effectively linked their conclusions to the original research question, ensuring that they are well-supported by the data presented. Overall, the validity of the findings is commendable, and they hold significant potential for further exploration and application in the literature.

Additional comments

Overall, the paper is well-prepared and presents a valuable contribution to the field. The primary area for improvement lies in enhancing the flow and expression of the language used throughout the manuscript.

Reviewer 2 ·

Basic reporting

Although the article is generally written in technical English, the abstract and introduction sections contain very long and congested sentences, which reduce readability.

The innovative aspects of the proposed SA3C-ID method can be clarified by providing more examples from the literature after 2020.

The datasets used (NSL-KDD, CSE-CIC-IDS2018) are known datasets in the field. However, the raw data, training/test separations, and preprocessing steps are not clearly stated. In addition, the implementation details of the SABPIO algorithm, hyperparameter values, and elements that affect reproducibility, such as training time, are missing.

Experimental design

1. Methodological Clarity
The details of the SABPIO algorithm are given quite superficially. Although the feature selection phase is very critical for the success of the system:
• The algorithmic structure of SABPIO is not clear (rule-based or ranking-based?).
• Why were other feature selection methods (Recursive Feature Elimination, PCA, SHAP) not preferred?
• The effect of feature selection on overall accuracy has not been shown experimentally.

2. Limited Representation of Adversarial Training
The model mentions adversarial training, however:
• What kind of attack does the “attacker” agent produce? Random noise or learned attack?
• How are the adversarial data samples generated?
• The effect of this training process on model performance has not been shown experimentally.

3. Imbalanced Datasets and Attack Types
Limited Discussion on Class Imbalance
• Especially in the NSL-KDD dataset, some attack types contain very few examples. This may cause the model to perform poorly in these minority classes. However, this analysis has not been performed.

Validity of the findings

The authors propose a new cyber attack detection model called SA3C-ID. This model aims to achieve more effective attack detection by combining components such as feature selection (SABPIO), adversarial training, and asynchronous actor-critic (A3C) reinforcement learning. The presented system has been tested on two different datasets (NSL-KDD and CSE-CIC-IDS2018) and has achieved promising results when compared to competing methods.

The paper presents an interesting and innovative approach, but requires significant improvements in methodology justification, experimental rigor, and presentation quality.

Additional comments

This work has the potential to make a significant contribution to the innovative use of artificial intelligence techniques in the detection of network attacks. However, due to the methodological uncertainties, lack of experimental validation, and inadequate explanations outlined above, it needs to be improved. The work could be significantly improved if the authors made the suggested corrections.

Reviewer 3 ·

Basic reporting

This paper proposed an adversarial intrusion detection model based on reinforcement learning (SA3C-ID). The impact of the paper must be made clear. The recent literature must be given. Some of the figures are not given in the necessary format.

Experimental design

The experimental part is given, but the discussion and conclusion sections must be given with more details, including the interpretations.

Validity of the findings

The findings are given well enough.

Additional comments

The impact of the paper must be made clear. The recent literature must be given. Some of the figures are not given in the necessary format.

·

Basic reporting

SA3C-ID like similar to other, adopting the SAC reinforcement learning algorithm for IDS. There are no new things. There are literature works, but they only show another method with results. No comparative study, not highlighting the disadvantages of other works, would like to solve or the advantage point adopted in this work. Unclear proposal description of the SA3C-ID Model section. It consists of a bad structure and a bad connection between sub-sections. There is no analysis of results with the IDS context. It seems individual case and individual methodology.

Experimental design

Experimental design only evaluates with two different datasets and evaluates them with general machine learning evaluation metrics. In this context, it must be correlated to the case context, which is IDS. Before that, an evaluation scenario with a clear setup must be described.

Validity of the findings

Since there is no correlation between case and methodology, result, and IDS context connection to problem and motivation, the validity of findings is questionable.

Additional comments

Please be better prepared, make a good connection between sentences, between paragraphs, between sections, also between problems, solutions, and results. Did the results closely relate to the problem of IDS you mentioned in the beginning?

Reviewer 5 ·

Basic reporting

The paper proposes SA3C-ID, an adversarial intrusion detection model leveraging reinforcement learning and advanced feature selection (SABPIO) for cybersecurity applications. The model incorporates asynchronous adversarial training between attacker and defender agents, models the detection task as a Markov decision process, and aims to improve detection performance on challenging, imbalanced network datasets (NSL-KDD and CSE-CIC-IDS2018). Results show competitive F1-scores compared to a suite of modern baselines.

Experimental design

- Important Topic
The work addresses pressing cybersecurity concerns, specifically intrusion detection in complex, modern network environments (cloud, IoT, 5G). The focus on imbalanced data and minority attack types is practically important.

- Clear Structure and Motivation
The paper gives a good background (feature selection, deep learning NIDS, RL NIDS), and explicitly lists the main contributions.

- Effective integration of adversarial training in a reinforcement learning (RL) framework.
The use of SABPIO for feature selection is novel and well-justified, aiming to reduce feature redundancy and improve efficiency.

Validity of the findings

- Thorough Experimental Evaluation
The model is tested on two standard, large-scale datasets: NSL-KDD and CSE-CIC-IDS2018.
Includes ablation studies to assess the contributions of each module (SABPIO, asynchronous training).
Comparative analysis against both classical and recent deep learning and RL-based models.

Additional comments

- Incremental Novelty
The adversarial RL approach, while interesting, draws upon existing adversarial reinforcement learning (AE-RL, AE-SAC) frameworks. The main innovation seems to be the combination with SABPIO and asynchronous training.
The originality is moderate, as several cited works (Caminero et al., Li et al., Tellache et al.) already apply adversarial or multi-agent RL to intrusion detection.

- Motivation Discussion
In section 2 related work, while discussing exiting works, please author discusses the limitations of them. And at the end of each category, you can explain how you method addresses these limitations.

---

## Round 0.2 · Minor Revisions

Reviewers have now commented on your paper. All concerns raised by the reviewers have been addressed; however, some parts require further work. This issue requires a minor revision. Please refer to the reviewers’ comments at the end of this letter; you will see that they advise you to revise your manuscript. If you are prepared to undertake the work required, I would be pleased to reconsider my decision. Please submit a list of changes or a rebuttal against each concern when you submit your revised manuscript.

Thank you for considering PeerJ Computer Science for the publication of your research.

With kind regards,

·

Basic reporting

authors have address all comments, but some parts only described at response letter. A critical information have to include and clearly describe at manuscript, like showing uniqueness, proposal description and more clear discussion. Besides, the conclusion section is takes too long paragraph.

Experimental design

authors have address comments, but only described at response letter.

Validity of the findings

Similarly, authors have address comments, but only described at response letter.

Additional comments

authors have address comments, but only at introduction section modified to improve the linking connection and structure. Other section structure must be adjust and improve.

---

## Round 0.3 · accepted · Accept

I am pleased to inform you that your work has now been accepted for publication in PeerJ Computer Science.

Please be advised that you cannot add or remove authors or references post-acceptance, regardless of the reviewers' request(s).

Thank you for submitting your work to this journal. I look forward to your continued contributions on behalf of the Editors of PeerJ Computer Science.

With kind regards,

·

Basic reporting

authors have address all comments.

Experimental design

authors have address all comments.

Validity of the findings

authors have address all comments.

Additional comments

authors have address all comments.